# Improving Visual Token Reduction via Rectifying Distortions for Efficient Multimodal LLM Inference

**Hyeonwoo Cho**[1]  **Donghyeon Baek**[1]  **Yewon Kim**[1]  **Bumsub Ham**[1 2]

## Abstract

Recent advancements in Multimodal Large Language Models (MLLMs) have achieved remarkable success in vision-language tasks, yet the quadratic computational complexity arising from the vast number of visual tokens incurs significant memory and latency bottlenecks. While visual token reduction (VTR) strategies have been explored to mitigate this burden, existing methods overlook the positional and attentional consistency between the full and reduced sequences, resulting in a distorted representation. To this end, we propose RESTORE, a novel VTR framework that rectifies the positional and attentional distortions while maintaining efficiency. Specifically, we present a simple yet effective calibration method that restores lost visual attention by augmenting attention weights based on relative distances. We also introduce a distinctive anchor selection for token merging to mitigate information loss during feature averaging. Experimental results on multiple benchmarks demonstrate that our method consistently improves the accuracy of various reduction methods, achieving state-of-the-art performance while maintaining computational efficiency. Project page is available at https://cvlab.yonsei.ac.kr/projects/RESTORE

## 1. Introduction

Recent advancements in Multimodal Large Language Models (MLLMs) (Liu et al., 2023; 2024a; Bai et al., 2025; Lin et al., 2024) have achieved remarkable success in interpreting complex visual information and translating it into coherent textual narratives, bridging the modality gap between vision and language. MLLMs leverage the generalization capabilities of Large Language Models (LLMs) (Radford et al., 2019; Brown et al., 2020; Touvron et al., 2023; Bai et al., 2023) with a pretrained visual encoder such as CLIP (Radford et al., 2021). An input image is projected into a sequence of visual tokens by the visual encoder, then processed via the LLM to generate text responses based on text inputs. Given that the computational complexity of the attention mechanism scales quadratically with sequence length, the vast number of visual tokens introduces a substantial bottleneck as they often reach into the thousands for high-resolution images or video inputs. Such an extensive visual sequence incurs significant memory and latency overheads.

To mitigate the computational burden, recent research has explored diverse strategies for visual token reduction (VTR) by pruning redundant tokens (Chen et al., 2024; Zhang et al., 2025b;a; Zou et al., 2025) or merging similar tokens (Bolya et al., 2023; Yang et al., 2025; Shang et al., 2025). Token pruning maintains the original information of retained tokens but sacrifices the context from pruned ones. Conversely, although token merging preserves global context by aggregating multiple tokens, averaging features during token merging leads to information loss of fine-grained details. A hybrid approach, VisionZip (Yang et al., 2025), attempts to combine the strengths of these approaches by first pruning and then merging the remaining tokens. Specifically, it preserves high-attention tokens and merges the rest into representatives (*i.e.*, anchor tokens) chosen via uniform sampling. However, the anchor tokens might fail to represent their groups, leading to significant information loss during the merging process.

Beyond the reduction strategies, we observe that existing VTR methods fail to preserve the total attention weights of visual tokens compared to the full token sequence. This attenuation stems from the normalization property of the softmax function. As the number of visual tokens decreases, the probabilities originally allocated to the reduced tokens are redistributed to the retained tokens. Due to the exponential nature of softmax, the redistribution amplifies the remaining tokens that possessed large attention weights. The loss of visual attention weights causes the model to neglect visual contexts and rely on textual information, po-

---

[1]Yonsei University [2]Korea Institute of Science and Technology (KIST). Correspondence to: Bumsub Ham <bumsub.ham@yonsei.ac.kr>.

*Proceedings of the 43rd International Conference on Machine Learning*, Seoul, South Korea. PMLR 306, 2026. Copyright 2026 by the author(s).

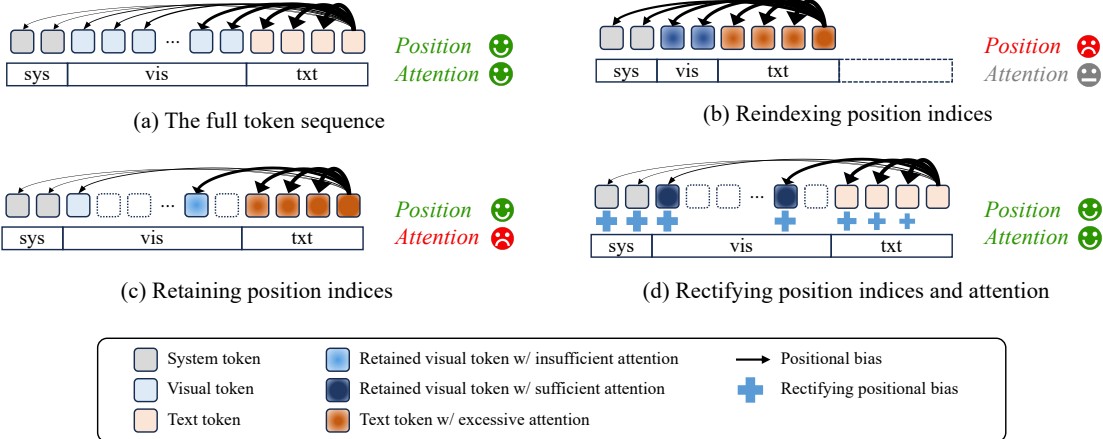

*Figure 1.* Illustration of the impact of visual token reduction on the internal attention mechanism of the LLM within MLLMs (e.g., LLaVA (Liu et al., 2024a)). (a) The full token sequence. (b) Reindexing position indices assigns contiguous indices to the reduced sequence. (c) Retaining position indices preserves the original indices of the retained tokens from (a). (d) We rectify distortions by retaining original position indices and calibrating the attention weights of the retained tokens.

tentially leading to weak visual grounding or hallucinations. We show in Fig. 1 the token sequences and their attention weights when the last text token serves as the query. Figure 1(a) illustrates the full token sequence, where attention is distributed across all tokens without reduction. For the reduced token sequence, existing methods either assign new contiguous position indices (Chen et al., 2024; Xing et al., 2024; Zhang et al., 2025a) (Fig. 1(b)) or retain the initial position indices from the full token sequence (Yang et al., 2025; Alvar et al., 2025; Zou et al., 2025) (Fig. 1(c)). The reindexing strategy reduces the relative distances between tokens, which helps to partially preserve the total attention weights of visual tokens. That is, squeezing the relative distances mitigates the positional bias of rotary position embedding (RoPE) (Su et al., 2024) in the original indices of the full sequence. However, this strategy disturbs spatial relationships between tokens. For instance, the substantial distance between the first visual token and the last text token is shortened, disrupting the spatial relationships compared to the full token sequence. This problem could be addressed by retaining original position indices, but at the cost of a significant drop in visual attention. Due to the positional bias, the attention weights originally assigned to reduced visual tokens are redistributed towards text tokens that are closer to the query.

In this paper, we propose **RESTORE**, a novel framework that **RE**ctifies di**S**tortions in visual **TO**ken **RE**duction. To address this, we introduce a calibration method that restores attention weights while preserving original position indices (Fig. 1(d)). This attention calibration counteracts the positional bias of RoPE, recovering the lost total attention weights of visual tokens. We also introduce a novel anchor

selection strategy for token merging that considers both representativeness and discriminativeness, minimizing information loss during feature averaging. Extensive experiments on several MLLM benchmarks demonstrate that our method consistently improves the accuracy of various VTR methods while maintaining efficiency. Our main contributions are summarized as follows:

- We analyze the attentional and positional distortions overlooked by existing VTR methods and propose a calibration method that rectifies the distortions.

- We introduce a distinctive anchor token selection strategy, mitigating information loss during token merging by selecting representative and discriminative anchor tokens.

- We provide comprehensive experimental results on multiple MLLM benchmarks, demonstrating that our method significantly improves accuracy across various VTR methods while preserving efficiency.

## 2. Related Work

**Text-aware visual token reduction.** Early VTR approaches for MLLMs focus on retaining visual tokens that are most relevant to the text input. The seminal work of FastV (Chen et al., 2024) introduces a text-aware token pruning method that leverages the cross-attention technique between visual and text tokens to identify and retain the most relevant visual tokens. Specifically, exploiting the autoregressive nature of LLMs, it selectively prunes visual tokens that exhibit low attention weights with respect to the last text token. FastV demonstrates that MLLMs maintain high accuracy even with a significantly reduced number of

visual tokens. Subsequent works build upon this idea with advanced mechanisms to assess token importance. Sparse-VLM (Zhang et al., 2025b) introduces a pruning metric that averages cross-attention weights towards a selected subset of text tokens that are highly correlated with the visual input. FitPrune (Ye et al., 2025) identifies the importance score of visual tokens by multiplying visual self-attention weights with cross-attention weights, considering visual saliency and text relevance jointly. While these methods achieve high accuracy with text-relevant visual information, they still require significant computation within the LLM layers to compute attention weights before reduction, which limits the efficiency.

**Text-agnostic visual token reduction.** Another line of research focuses on reducing visual tokens without considering text relevance. These approaches reduce visual tokens before feeding the tokens into the LLM, avoiding the computational overhead of text-aware methods. They prune or merge redundant tokens based on attention weights relative to the [CLS] token for measuring importance or a self-correlation of visual features for measuring similarity. For pruning, VisPruner (Zhang et al., 2025a) incorporates both attention weights and feature similarity for token selection, while DART (Wen et al., 2025) leverages only feature similarity to eliminate redundancy. DivPrune (Alvar et al., 2025) emphasizes diversity between retained tokens by selecting a subset of tokens that are mutually distant. Our distinctive anchor token selection shares a similar motivation, but distinguishes itself in the criteria for token merging. To be specific, we incorporate the correlation between anchors and remaining tokens to mitigate information loss during feature averaging. HoloV (Zou et al., 2025) introduces a dynamic scoring based on feature variance and attention saliency to adaptively allocate the token budget across spatial partitions. Token pruning ensures the integrity of the retained tokens, but it discards information from pruned tokens. This exclusion often leads to the loss of text-relevant visual details, causing performance degradation. Merging-based approaches have been proposed to preserve information by aggregating multiple tokens into a representative anchor token. These methods primarily diverge in their strategies for selecting anchor tokens. PruMerge (Shang et al., 2025) selects anchor tokens with high attention weights, while VisionZip (Yang et al., 2025) performs merging with uniformly distributed anchor tokens. These strategies are likely to select suboptimal anchor tokens since they exhibit low similarity to the tokens they merge. Consequently, the anchor tokens fail to represent their local neighborhoods, exacerbating information loss during feature averaging. To overcome this limitation, we introduce a distinctive anchor selection for token merging that prioritizes tokens with high representativeness for their groups and discriminativeness from other anchor tokens.

Furthermore, we identify and address the overlooked issue of attentional distortion in existing VTR methods. Existing approaches differ in their position assignment to the reduced token sequence, exploiting either reindexing (Chen et al., 2024; Xing et al., 2024; Zhang et al., 2025a) or retaining original indices (Yang et al., 2025; Alvar et al., 2025; Zou et al., 2025). However, the impact of these position assignments on attention weights and subsequent model performance remains unexplored. To the best of our knowledge, we are the first to provide an in-depth analysis of the position assignment for the reduced token sequence. Building on the analysis, we propose a novel method to calibrate attention distortion.

## 3. Method

In this section, we first describe the MLLM architecture briefly (Sec. 3.1). We then provide a detailed description of our framework with an in-depth analysis (Sec. 3.2), and introduce a distinctive anchor selection for token merging (Sec. 3.3).

### 3.1. Preliminaries

**VTR in MLLMs.** We adopt the architecture of standard MLLMs, such as LLaVA (Liu et al., 2023), which comprises a visual encoder, a projector, and an LLM. Given an input image, the visual encoder extracts visual features, which are then mapped into the LLM's embedding space via the projector to obtain a sequence of visual tokens $\mathbf{X}_{\text{vis}} \in \mathbb{R}^{N_{\text{vis}} \times d}$, where $N_{\text{vis}}$ denotes the original number of visual tokens and $d$ is the hidden dimension. To mitigate computational overhead, the VTR method is applied to $\mathbf{X}_{\text{vis}}$ before feeding it into the LLM. This process transforms the original sequence into a reduced set of visual tokens $\hat{\mathbf{X}}_{\text{vis}} \in \mathbb{R}^{n_{\text{vis}} \times d}$, where $n_{\text{vis}} \ll N_{\text{vis}}$. The reduced visual tokens are then concatenated with system tokens $\mathbf{X}_{\text{sys}} \in \mathbb{R}^{N_{\text{sys}} \times d}$ and text tokens $\mathbf{X}_{\text{txt}} \in \mathbb{R}^{N_{\text{txt}} \times d}$ to form the token sequence $\mathbf{X} = [\mathbf{X}_{\text{sys}}; \hat{\mathbf{X}}_{\text{vis}}; \mathbf{X}_{\text{txt}}] \in \mathbb{R}^{N \times d}$, where $N = N_{\text{sys}} + n_{\text{vis}} + N_{\text{txt}}$. This concatenated sequence $\mathbf{X}$ serves as the input to the LLM for response generation.

**Self-attention with RoPE.** Modern LLMs primarily employ RoPE (Su et al., 2024) to encode spatial relationships between tokens. RoPE groups adjacent pairs of the feature dimension to form complex numbers, and then applies a rotation in the complex domain based on the token's position index. Let $\mathbf{x}_m, \mathbf{x}_n \in \mathbb{R}^{d_h}$ denote the feature vectors of the tokens at position $m$ and $n$ within the sequence $\mathbf{X}$, respectively, where $d_h$ denotes the hidden dimension of each head in the multi-head attention (Vaswani et al., 2017). Given the frequency parameters $\Theta = \{\theta_j = 10000^{-2(j-1)/d_h} \mid j \in [1, \ldots, d_h/2]\}$, the query and key vectors for tokens at positions $m$ and $n$ are as follows:

$$\mathbf{q}_m = \mathbf{W}_q(\mathbf{x}_m)e^{im\Theta}, \quad \mathbf{k}_n = \mathbf{W}_k(\mathbf{x}_n)e^{in\Theta}, \quad (1)$$

where $\mathbf{W_q}$ and $\mathbf{W_k}$ are the projection matrices for the query and key in the complex domain, respectively. A logit of attention weight $z_{m,n}$ is then computed as the real part of the inner product between the complex query and key vectors as follows:

$$z_{m,n} = \frac{\text{Re}\left(\sum_{j=1}^{d_h/2} \mathbf{W}_q(\mathbf{x}_m)_j \mathbf{W}_k(\mathbf{x}_n)_j^* e^{i|m-n|\theta_j}\right)}{\sqrt{d_h}}. \quad (2)$$

This formulation demonstrates that the attention weights depend on the relative distance $|m-n|$ through the phase shift $e^{i|m-n|\theta_j}$, enabling the attention mechanism to capture spatial relationships. However, VTR disrupts this mechanism by reducing tokens or altering positional indices, which distorts attentional or positional information. In the following section, we analyze these distortions and propose a method to rectify them.

### 3.2. Rectifying Distortions

**Analysis.** To analyze the impact of position assignments on the attention mechanism within the LLM, we compute the proportion of attention weights allocated to visual tokens averaged across all heads in each layer of LLaVA-1.5 (Liu et al., 2024a). That is, we compare the ratio of the original total attention weights of visual tokens that the reduced visual tokens retain. We measure this ratio for two distinct scenarios: when the query is a visual token (Visual-to-Visual) and when it is a text token (Text-to-Visual). We conduct a comparative analysis using a pruning method of HoloV (Zou et al., 2025) on the GQA (Hudson & Manning, 2019) dataset under four settings: the baseline using the full token sequence (Fig. 1(a), $N_{vis} = 576$), and reduced sequences ($n_{vis} = 64$) with reindexing indices (Fig. 1(b)), retaining original indices (Fig. 1(c)), and our proposed method (Fig. 1(d)). We show in Fig. 2(a) and Fig. 2(b) visual-to-visual and text-to-visual scenarios, respectively. These figures demonstrate that neither position assignment preserves the total attention weights of visual tokens at the baseline level. The reindexing strategy (dashed red line) exhibits a decline in attention weights, while retaining the original position indices (dashed green line) leads to a further substantial drop.

The attention attenuation of visual tokens is attributed to two primary reasons. First, the total attention weight of visual tokens inevitably diminishes in proportion to the reduced token count. Second, the further degradation observed in the retaining position indices arises from the positional bias of RoPE, where tokens with larger relative distances are penalized. This attenuation compromises the model's ability

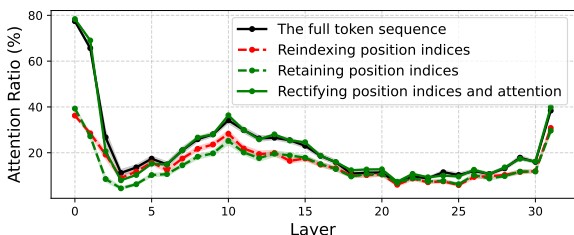

*(a)* Visual-to-visual attention.

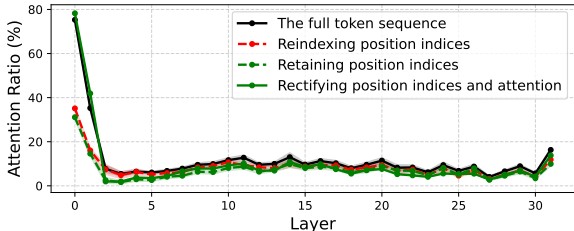

*(b)* Text-to-visual attention.

*Figure 2.* Comparison of average attention proportions assigned to visual tokens within the LLM (a) when the query is a visual token and (b) when the query is a text token.

to attend to visual information, resulting in weakened visual grounding and hallucinations. Therefore, it is essential for visual tokens to restore the attention weights of the reduced tokens to a level comparable to that of the full token sequence. To address these, we propose to calibrate attention weights, recovering the lost visual attention weights caused by the reduced token sequence and the long-term decay of RoPE.

**Attention calibration.** To address the attention attenuation while preserving spatial relationships, we first measure the impact of the long-term decay of RoPE quantitatively. We derive this decay through the isolation of the positional encoding component from Eq. 2. Specifically, extracting the real part of the rotation components and averaging them across the feature dimension $d_h$, we quantify how the attention weight attenuates as a function of the relative distance $|m-n|$. This yields the decay function $\mathcal{D}(|m-n|)$ as:

$$\mathcal{D}(|m-n|) = \frac{2}{d_h} \sum_{j=1}^{d_h/2} \cos\left(|m-n|\theta_j\right). \quad (3)$$

This function demonstrates that attention weights diminish with an oscillation as $|m-n|$ increases (Fig. 3 (blue)). When retaining original indices, the sparse visual tokens maintain large relative distances from the text query, causing their attention weights to be heavily penalized by the decay $\mathcal{D}$. On the other hand, text tokens remain adjacent to each other, maintaining relatively high values of $\mathcal{D}$. This disparity is

further exacerbated by the exponential nature of the softmax function in the reduced sequence. Since text tokens possess larger softmax logits due to their close distance, the softmax redistributes the probability toward them. Consequently, the sparse visual tokens fail to preserve the total attention weights originally allocated to the visual tokens in the full sequence. This attenuation compels the model to neglect visual contexts and rely on textual information. To rectify this, we introduce a distance-aware calibration term to the attention weights. The goal is to compensate for the decay $\mathcal{D}(|m - n|)$, leading distant visual tokens to regain their significance. Incorporating the token size $s_n$ for merging, the calibrated attention $\hat{A}_{m,n}$ is formulated as:

$$\hat{A}_{m,n} = \frac{\exp\left(z_{m,n} + \log s_n (c - \mathcal{D}(|m - n|))\right)}{\sum_{i=1}^{N} \exp\left(z_{m,i} + \log s_i (c - \mathcal{D}(|m - i|))\right)}, \quad (4)$$

where $s_n$ denotes the number of tokens that have been merged into the $n$-th token, and $c$ is a constant ensuring the rectification term remains positive. The logarithmic term $\log s_n$ ensures that a merged token receives weight proportional to the number of original tokens it represents, a technique introduced in ToMe (Bolya et al., 2023). In the presence of the positional bias induced by RoPE, attention weights are shifted toward text tokens as the visual sequence becomes sparse. Thus relying solely on $s_n$ fails to prevent the suppression of visual context. To address this, we augment $s_n$ with the distance-aware component $(c - \mathcal{D}(|m - n|))$ to counteract the positional bias. The distance-aware term acts as a counterweight to the RoPE decay (Fig. 3 (orange)). This restores the magnitude of attention weights for distant visual tokens, aligning the distribution with that of the full token sequence as shown in Fig. 2 (solid green line). Note that while $\log s_n$ was successfully employed in ToMe for visual-only tasks, we argue that it is insufficient for MLLMs due to the interference of textual tokens.

Furthermore, while our derivation primarily focuses on standard 1D RoPE, the fundamental principle of distance-aware calibration seamlessly generalizes to multi-dimensional spatial encodings, such as the multimodal RoPE (M-RoPE) employed in recent models (*e.g.*, Qwen2.5-VL (Bai et al., 2025)). We provide the theoretical generalization (Appendix A) and empirical validation for M-RoPE (Sec. 4).

### 3.3. Distinctive Anchor Token Selection

To minimize information loss during the merging process, it is crucial to select anchor tokens that are not only informative but also mutually exclusive to avoid redundancy. The previous methods rely on uniform sampling (Yang et al., 2025) or high-attention tokens (Shang et al., 2025) for selecting anchor tokens. However, these approaches suffer

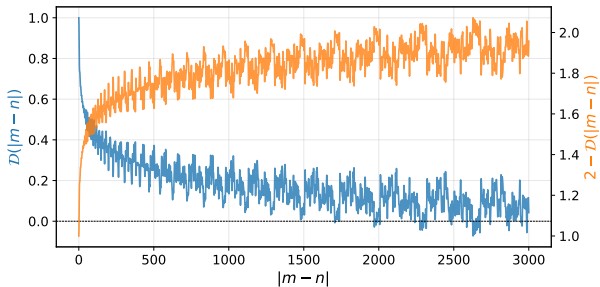

*Figure 3.* Visualizations of the long-term decay $\mathcal{D}(|m - n|)$ (blue) and our calibration term (orange). The calibration term increases with relative distance to counteract the long-term decay.

from two limitations: (1) Selecting anchor tokens with low correlation to non-anchor tokens results in poor representativeness. In this scenario, the anchor token fails to serve as a centroid for its cluster, exacerbating information loss during merging stage. (2) Selecting multiple anchor tokens that are highly correlated to each other introduces redundancy, as these tokens occupy overlapping regions in feature space and should ideally be merged rather than preserved individually. To address these, inspired by density peak clustering (Rodriguez & Laio, 2014), we propose an anchor token selection strategy based on feature similarity, guided by two criteria: representativeness that ensures anchor tokens are highly correlated to other tokens, and discriminativeness that prevents redundancy among selected anchors. We first define the representativeness $\mathcal{R}$ as the sum of pairwise correlations $\mathbf{C}_{ij}$ with other tokens as follows:

$$\mathcal{R}_i = \sum_{j=1}^{N_{\text{vis}}} \mathbf{C}_{ij}, \quad \text{where} \quad \mathbf{C} = \mathbf{X}_{\text{vis}} \mathbf{X}_{\text{vis}}^T / \|\mathbf{X}_{\text{vis}}\|^2. \quad (5)$$

$\mathcal{R}_i$ quantifies the accumulated similarity of the $i$-th token to all other visual tokens, indicating its potential to serve as an anchor token. However, relying solely on $\mathcal{R}_i$ may lead to redundant anchor selection if candidates are highly correlated to each other. To identify and suppress such redundancy, we introduce a binary mask $\mathbf{M}$ that flags superior tokens (*i.e.*, tokens with higher representativeness) as follows:

$$\mathbf{M}_{ij} = \mathbb{I}(\mathcal{R}_j > \mathcal{R}_i). \quad (6)$$

Subsequently, we apply this mask to $\mathbf{C}$ to isolate correlation solely towards superior counterparts. The masked correlation matrix $\hat{\mathbf{C}}$ is defined as:

$$\hat{\mathbf{C}}_{ij} = \begin{cases} \mathbf{C}_{ij} & \text{if } \mathbf{M}_{ij} = 1 \\ -\infty & \text{otherwise} \end{cases} . \quad (7)$$

Based on $\hat{\mathbf{C}}$, we measure the redundancy of the $i$-th token by finding its maximum correlation with any superior token ($\max_j \hat{\mathbf{C}}_{ij}$). A high maximum value implies that the token is substantially covered by a more representative token. Conversely, a low value indicates that the token captures unique features. We therefore define the discriminativeness as $1 - \max_j \hat{\mathbf{C}}_{ij}$. Finally, the anchor set $\mathcal{A}$ is selected by weighting the representativeness with discriminativeness as follows:

$$\mathcal{A} = \text{Top-K}\left(\mathcal{R}_i \odot \left(1 - \max_j \hat{\mathbf{C}}_{ij}\right)\right), \qquad (8)$$

where $\text{Top-K}(\cdot)$ selects the top-$K$ tokens, and $\odot$ denotes element-wise multiplication. With the optimal anchor set $\mathcal{A}$, the other tokens are merged into the anchor tokens with which they share the highest correlation. Note that this process incurs no additional computational overhead, as we reuse the pairwise correlation values from the pre-computed correlation matrix $\mathbf{C}$.

## 4. Experiments

In this section, we describe implementation details (Sec. 4.1), and present quantitative comparisons between previous VTR methods and our method (Sec. 4.2). We then conduct a detailed analysis of our framework (Sec. 4.3). Additional results and discussions are provided in the Appendix.

### 4.1. Implementation details

**Datasets and models.** We evaluate our method on multiple MLLM benchmarks, including GQA (Hudson & Manning, 2019), MMBench (Liu et al., 2024b), MME (Fu et al., 2025), POPE (Li et al., 2023), Science-QA (SQA) (Lu et al., 2022), VQAv2 (Goyal et al., 2017), TextVQA (Singh et al., 2019), SEED-Bench (Li et al., 2024) datasets for image question answering tasks. We perform experiments using LLaVA-1.5-7B, LLaVA-NeXT-7B (Liu et al., 2024a), and Qwen2.5-VL-7B (Bai et al., 2025).

**Baselines.** We compare several VTR approaches including text-aware pruning methods such as FastV (Chen et al., 2024), PDrop (Xing et al., 2024), SparseVLM (Zhang et al., 2025b), and text-agnostic methods such as VisionZip (Yang et al., 2025), DivPrune (Alvar et al., 2025), VisPruner (Zhang et al., 2025a), HoloV (Zou et al., 2025). We integrate our framework with the text-agnostic methods for evaluation, as they are more efficient than text-aware methods. Following the protocol of VisionZip, we adopt a hybrid reduction strategy that sequentially applies pruning and merging to integrate our merging method. Specifically, we first prune $\gamma n_{vis}$ tokens based on the pruning criteria of

the four baselines, and then merge the remaining $(1-\gamma)n_{vis}$ tokens. Unless otherwise specified, we set the pruning ratio to $\gamma = 0.5$ (*i.e.*, the token budget is equally distributed between the pruning and merging stages). For the hyperparameter $c$, we set $c = 2$ by default, ensuring the minimum value of the calibration term is 1.

### 4.2. Results

We show in Table 1 the results on eight benchmarks across three different token retaining ratios. The experimental results demonstrate that our proposed framework consistently enhances the performance of various text-agnostic VTR baselines, including VisionZip, DivPrune, VisPruner, and HoloV. When retaining 192 or 128 tokens, our method achieves nearly lossless performance compared to the vanilla baseline. For instance, when integrated with VisPruner and HoloV at a retention of 192 tokens, our approach maintains over 98% of the original performance. For an aggressive reduction to 64 tokens, our method with HoloV achieves an average accuracy of 96.5%, significantly outperforming the baseline HoloV by 4.3%. This indicates that our method effectively rectifies distortions and preserves visual information even under challenging reduction scenarios. We also show in Table 2 the results of LLaVA-NeXT-7B on multiple benchmarks. Given that this model processes extensive visual sequences, it inherently contains significant redundancy. Our method consistently achieves performance improvements across methods and retention ratios, demonstrating the versatility of our approach across different MLLM architectures. We further evaluate our framework on Qwen2.5-VL-7B-Instruct (Bai et al., 2025), which leverages M-RoPE. Since our distance-aware calibration naturally generalizes to such multi-dimensional encodings (Appendix A), it remains directly applicable. As shown in Table 3, our method consistently improves all baselines across every retention ratio.

### 4.3. Discussions

**Ablations.** We show in Table 4 an ablation study on the components of our framework. Unless otherwise specified, all studies in this section report accuracy averaged over the eight benchmarks in Table 1 under the aggressive retention ratio of $n_{vis} = 64$, where each component is evaluated by isolating its effect while fixing the others. We adopt this challenging ratio to make the contribution of each component clearly observable, and use HoloV as the base method, which is the most recent baseline. The baseline ① represents HoloV using only pruning. The results in rows ② and ③ show that applying either retaining position indices or attention calibration leads to performance degradation compared to the baseline. Specifically, retaining original indices without attention calibration (③) drops the accuracy from 92.2% to 90.6%. This supports our analysis that

*Table 1.* Comparison with VTR methods for LLaVA-1.5-7B (Liu et al., 2024a) on eight benchmarks and the average score across them. For baselines, all experiment results are re-implemented from their official codebases under the same environments. Best results at each reduction ratio in bold.

| Type | Method | Average | GQA | MMB | MME | POPE | SQA$^{IMG}$ | VQA$^{V2}$ | VQA$^{Text}$ | SEED |
|------|--------|---------|-----|-----|-----|------|------|------|------|------|
| | | *Using All Visual Tokens, 576 Tokens (100%)* | | | | | | | | |
| - | LLaVA-1.5-7B | 100% | 61.9 | 64.6 | 1862 | 85.9 | 69.5 | 78.5 | 58.2 | 58.6 |
| | | *Retain 192 Tokens (33.3%)* | | | | | | | | |
| Text-aware | FastV (ECCV24) | 96.0% | 57.1 | 64.4 | 1821 | 75.8 | 68.9 | 74.7 | 57.8 | 56.3 |
| | PDrop (CVPR25) | 96.8% | 58.0 | 62.9 | 1790 | 84.0 | 68.9 | 76.0 | 57.1 | 55.8 |
| | SparseVLM (ICML25) | 98.1% | 59.5 | 64.2 | 1782 | 85.4 | 68.7 | 77.0 | 57.7 | 57.3 |
| Text-agnostic | ToMe (ICLR23) | 96.9% | 59.5 | 62.6 | 1727 | 86.9 | 69.0 | 75.9 | 55.8 | 56.6 |
| | VisionZip (CVPR25) | 96.8% | 59.2 | 62.5 | 1749 | 85.2 | 68.7 | 77.2 | 55.8 | 56.3 |
| | + ours | 98.0% | 60.6 | 63.7 | 1782 | 86.6 | 69.1 | 77.0 | 54.9 | 58.0 |
| | DivPrune (CVPR25) | 96.9% | 58.9 | 63.1 | 1723 | 86.5 | 69.0 | 76.1 | 55.7 | 56.8 |
| | + ours | 98.7% | 60.9 | 63.7 | 1813 | 86.6 | 69.1 | 77.4 | 56.5 | 58.2 |
| | VisPruner (ICCV25) | 97.5% | 59.4 | 62.5 | 1784 | 86.0 | 68.3 | 76.5 | 57.7 | 56.6 |
| | + ours | **98.8%** | 60.9 | 63.3 | 1816 | 86.1 | 69.5 | 77.6 | 57.0 | 58.1 |
| | HoloV (NeurIPS25) | 96.5% | 58.6 | 62.6 | 1779 | 85.0 | 67.3 | 76.0 | 55.8 | 56.3 |
| | + ours | **98.8%** | 61.0 | 63.7 | 1793 | 86.6 | 69.6 | 77.6 | 57.2 | 58.1 |
| | | *Retain 128 Tokens (22.2%)* | | | | | | | | |
| Text-aware | FastV (ECCV24) | 91.8% | 54.1 | 63.1 | 1694 | 68.3 | 69.3 | 70.7 | 56.3 | 54.0 |
| | PDrop (CVPR25) | 84.5% | 51.6 | 54.9 | 1403 | 67.8 | 68.8 | 65.6 | 52.6 | 47.0 |
| | SparseVLM (ICML25) | 97.2% | 58.4 | 64.4 | 1761 | 85.0 | 68.7 | 76.2 | 56.6 | 56.9 |
| Text-agnostic | ToMe (ICLR23) | 95.1% | 58.7 | 60.7 | 1668 | 86.5 | 68.4 | 74.5 | 54.8 | 55.1 |
| | VisionZip (CVPR25) | 95.6% | 58.5 | 61.4 | 1705 | 83.2 | 68.8 | 76.7 | 55.6 | 55.2 |
| | + ours | 97.1% | 59.8 | 62.7 | 1772 | 86.0 | 68.4 | 76.6 | 55.0 | 57.2 |
| | DivPrune (CVPR25) | 96.3% | 58.6 | 63.7 | 1702 | 86.5 | 68.9 | 75.2 | 55.2 | 55.8 |
| | + ours | 97.9% | 60.6 | 63.1 | 1795 | 86.0 | 68.6 | 76.8 | 56.0 | 57.8 |
| | VisPruner (ICCV25) | 96.3% | 58.1 | 61.9 | 1778 | 84.5 | 68.8 | 75.3 | 56.9 | 55.0 |
| | + ours | **98.4%** | 60.9 | 63.3 | 1813 | 86.2 | 68.7 | 77.1 | 56.4 | 57.9 |
| | HoloV (NeurIPS25) | 95.5% | 57.5 | 62.5 | 1761 | 82.2 | 69.0 | 75.0 | 55.6 | 54.7 |
| | + ours | 98.3% | 60.8 | 63.0 | 1807 | 86.0 | 68.7 | 77.1 | 56.6 | 58.0 |
| | | *Retain 64 Tokens (11.1%)* | | | | | | | | |
| Text-aware | FastV (ECCV24) | 75.2% | 46.4 | 51.4 | 1284 | 36.1 | 69.9 | 56.2 | 51.6 | 43.9 |
| | PDrop (CVPR25) | 80.9% | 49.0 | 55.6 | 1404 | 55.5 | 69.5 | 60.1 | 50.6 | 46.1 |
| | SparseVLM (ICML25) | 90.6% | 53.8 | 60.2 | 1591 | 77.5 | 69.7 | 70.3 | 53.5 | 51.2 |
| Text-agnostic | ToMe (ICLR23) | 91.6% | 56.0 | 57.9 | 1588 | 84.3 | 68.0 | 71.7 | 52.6 | 52.8 |
| | VisionZip (CVPR25) | 93.0% | 56.0 | 60.5 | 1690 | 78.2 | 69.5 | 75.2 | 53.8 | 52.8 |
| | + ours | 94.8% | 58.5 | 62.3 | 1726 | 84.6 | 68.0 | 74.3 | 52.2 | 55.2 |
| | DivPrune (CVPR25) | 93.8% | 57.1 | 60.2 | 1653 | 85.3 | 68.3 | 73.3 | 54.5 | 53.7 |
| | + ours | 95.9% | 59.0 | 62.2 | 1748 | 84.8 | 68.0 | 75.0 | 54.4 | 56.3 |
| | VisPruner (ICCV25) | 92.8% | 55.8 | 60.0 | 1670 | 80.5 | 68.5 | 72.3 | 55.6 | 52.6 |
| | + ours | 96.0% | 59.2 | 61.6 | 1722 | 85.1 | 68.0 | 75.4 | 55.4 | 56.5 |
| | HoloV (NeurIPS25) | 92.2% | 55.1 | 60.0 | 1699 | 76.8 | 68.6 | 72.3 | 54.9 | 52.5 |
| | + ours | **96.5%** | 59.0 | 61.9 | 1787 | 84.9 | 68.0 | 75.6 | 55.4 | 56.7 |

sparse positional indices induce severe attention decay due to the long-term property of RoPE, causing the model to neglect visual tokens, which is consistent with the analysis of Fig. 2. Applying the attention calibration technique alone (②) also results in a performance drop to 91.0%. The simultaneous deployment of both strategies (row ④) yields a performance gain, confirming their complementary nature in resolving positional and attentional distortions. Comparing the pruning-only results ① with the merging ⑤, we observe that merging improves performance from 92.2% to 93.5%. This confirms that aggregating information into distinctive anchors is more effective than discarding tokens. In contrast, augmenting merging with attention calibration under reindexed positions (row ⑥) slightly lowers the accuracy to 92.9%, as calibrating reindexed sequences amplifies

attention to visual tokens excessively, which shares the same mechanism as the drop in row ② (see Fig. 5 in Appendix D). Finally, the full framework in row ⑧ achieves the highest accuracy of 96.5%, demonstrating the synergistic effect of integrating merging with calibration. A notable comparison arises between ⑦ and ⑧. In row ⑦, applying merging with only position rectification results in the lowest performance of 90.1%. This arises because retaining original indices places merged tokens at large relative distances from the query. As a result, RoPE suppresses these semantically rich tokens, preventing the model from attending to the aggregated global context. We address this via attention calibration, leading to a substantial improvement of 6.4%.

*Table 2.* Comparison with VTR methods on LLaVA-NeXT-7B (Liu et al., 2024a). Best results at each reduction ratio in bold.

| Method | Average | GQA | POPE | VQA$^{\text{Text}}$ | SEED |
|---|---|---|---|---|---|
| *Using All Visual Tokens, 2880 Tokens* **(100%)** | | | | | |
| LLaVA-Next-7B | 100% | 64.2 | 86.5 | 64.9 | 70.2 |
| *Retain 320 Tokens* **(11.1%)** | | | | | |
| Visionzip (CVPR25) | 88.9% | 59.0 | 82.7 | 55.2 | 58.3 |
| + ours | 90.5% | 60.6 | 87.2 | 53.0 | 59.9 |
| DivPrune (CVPR25) | 88.8% | 60.2 | 84.0 | 51.6 | 59.5 |
| + ours | 90.2% | 60.7 | 87.3 | 51.4 | 60.5 |
| VisPruner (ICCV25) | 90.8% | 59.3 | 83.2 | 59.1 | 58.6 |
| + ours | 91.9% | 60.9 | 87.2 | 55.9 | 60.2 |
| HoloV (NeurIPS25) | 87.5% | 59.6 | 83.4 | 57.1 | 51.0 |
| + ours | **92.0%** | 61.0 | 87.4 | 56.1 | 60.0 |
| *Retain 160 Tokens* **(5.5%)** | | | | | |
| Visionzip (CVPR25) | 84.9% | 56.2 | 77.6 | 54.5 | 55.0 |
| + ours | 87.9% | 58.5 | 85.5 | 51.2 | 58.0 |
| DivPrune (CVPR25) | 86.3% | 58.4 | 81.3 | 51.2 | 57.2 |
| + ours | 87.2% | 59.0 | 86.4 | 48.3 | 58.0 |
| VisPruner (ICCV25) | 86.7% | 57.0 | 78.6 | 57.1 | 55.6 |
| + ours | 88.5% | 58.8 | 85.9 | 52.0 | 58.2 |
| HoloV (NeurIPS25) | 86.3% | 57.2 | 78.1 | 56.0 | 55.9 |
| + ours | **89.0%** | 59.2 | 86.1 | 53.1 | 57.8 |

*Table 3.* Comparison with VTR methods on Qwen2.5-VL-7B-Instruct (Bai et al., 2025). Best results at each reduction ratio in bold.

| Method | Average | MMB | MME | POPE | SQA | TextVQA |
|---|---|---|---|---|---|---|
| *Using All Visual Tokens* | | | | | | |
| Qwen2.5-VL-7B-Instruct | 100% | 84.4 | 2323 | 86.7 | 77.8 | 77.7 |
| *Retain 33.3% visual tokens* | | | | | | |
| DivPrune (CVPR 25) | 95.9% | 79.6 | 2187 | 84.9 | 76.2 | 73.8 |
| +ours | 97.2% | 81.1 | 2205 | 86.2 | 78.2 | 73.8 |
| VisPruner (ICCV 25) | 95.8% | 80.2 | 2175 | 84.6 | 76.7 | 73.2 |
| +ours | **97.4%** | 82.0 | 2211 | 86.0 | 78.6 | 73.3 |
| HoloV (NeurIPS 25) | 93.6% | 78.2 | 2121 | 83.6 | 75.9 | 70.0 |
| +ours | 96.4% | 81.0 | 2193 | 85.7 | 78.1 | 71.9 |
| *Retain 22.2% visual tokens* | | | | | | |
| DivPrune (CVPR 25) | 93.9% | 77.6 | 2137 | 84.1 | 75.0 | 71.7 |
| +ours | **95.2%** | 79.8 | 2166 | 84.8 | 77.3 | 70.6 |
| VisPruner (ICCV 25) | 88.6% | 78.0 | 1604 | 82.4 | 75.7 | 69.2 |
| +ours | 94.8% | 79.0 | 2137 | 84.6 | 78.4 | 69.8 |
| HoloV (NeurIPS 25) | 90.6% | 76.5 | 2031 | 81.9 | 74.4 | 65.9 |
| +ours | 93.9% | 79.3 | 2166 | 84.4 | 76.6 | 67.4 |
| *Retain 11.1% visual tokens* | | | | | | |
| DivPrune (CVPR 25) | 87.5% | 71.1 | 1961 | 79.7 | 72.7 | 64.9 |
| +ours | **89.8%** | 74.5 | 1998 | 82.3 | 76.3 | 63.6 |
| VisPruner (ICCV 25) | 86.9% | 69.8 | 2137 | 77.0 | 73.1 | 59.8 |
| +ours | 88.5% | 74.4 | 1980 | 80.8 | 77.1 | 59.6 |
| HoloV (NeurIPS 25) | 83.5% | 72.3 | 1834 | 77.6 | 70.8 | 56.4 |
| +ours | 88.4% | 76.3 | 1992 | 81.3 | 75.4 | 58.3 |

**Impact of calibration terms.** We show in Table 5 the contribution of each term in our attention calibration. The baseline (w/o Attention Calibration) denotes that original position indices are retained but no attention calibration is applied to Eq. 4. Incorporating the merged group size term $s_n$ yields an improvement across all methods, recovering the attention weights proportional to the number of merged tokens. The best performance is consistently achieved when the distance-aware term is weighted. This confirms that scaling by merged group size is insufficient, demonstrating that it is crucial to compensate for the positional bias to recover lost attention weights for visual tokens.

**Effectiveness of distinctive anchor selection.** Table 6 compares different reduction types: pruning only, hybrid reduction with uniformly distributed anchor selection, and

*Table 4.* Ablation study on the components of our framework.

| | Incorporating Merging | Retaining Position | Attention Calibration | Average |
|---|---|---|---|---|
| ① | ✗ | ✗ | ✗ | 92.2% |
| ② | ✗ | ✗ | ✓ | 91.0% |
| ③ | ✗ | ✓ | ✗ | 90.6% |
| ④ | ✗ | ✓ | ✓ | 93.1% |
| ⑤ | ✓ | ✗ | ✗ | 93.5% |
| ⑥ | ✓ | ✗ | ✓ | 92.9% |
| ⑦ | ✓ | ✓ | ✗ | 90.1% |
| ⑧ | ✓ | ✓ | ✓ | **96.5%** |

*Table 5.* Comparison with attention calibration strategies.

| Method | w/o Attention Calibration | $s_n$ | $s_n(c - \mathcal{D}(|m - n|))$ |
|---|---|---|---|
| VisionZip | 89.5% | 94.4% | **94.8%** |
| DivPrune | 88.9% | 94.9% | **95.9%** |
| VisPruner | 88.7% | 95.1% | **96.0%** |
| HoloV | 90.1% | 95.3% | **96.5%** |

hybrid reduction with our distinctive merging. While the hybrid approach generally outperforms pruning-only methods, uniformly distributed anchor selection shows suboptimal performance. For instance, in the case of DivPrune, employing uniform anchor selection degrades performance compared to pruning (95.8% → 94.9%), suggesting that aggregating tokens into unrepresentative anchors causes information distortion. In contrast, our distinctive anchor selection strategy consistently outperforms the pruning-only counterpart across all baselines.

**Position of merged tokens.** Since a merged token serves as a representative for multiple tokens, assigning an appropriate positional index to this aggregate token is critical for preserving spatial context. We investigate four strategies for defining the position of the merged token: namely First, Median, Mean, and Last. These correspond to assigning the index of the earliest, central, arithmetic mean, and latest token within the group, respectively. As shown in Table 7, the 'Median' yields superior accuracy (96.5%). We conjecture that the median position offers the most accurate spatial approximation of the group's center, minimizing spatial distortion. Conversely, the 'Last' strategy results in the lowest accuracy (95.2%). This is because our attention calibration assigns a smaller boosting weight to positions closer to the text query (*i.e.*, the 'Last' position), which may result in insufficient attention to the merged token.

**Efficiency analysis.** The primary source of additional computational overhead in our framework arises from the anchor selection process. Specifically, computing the self-correlation matrix involves matrix multiplication with a complexity of $O(N_{vis}^2 d)$. However, this overhead is negli-

*Table 6.* Comparison with reduction types. P denotes pruning and M denotes merging.

| Method | P | P + M (uniform) | P + M (distinctive) |
|---|---|---|---|
| VisionZip | - | 94.4% | **94.8%** |
| DivPrune | 95.8% | 94.9% | **95.9%** |
| VisPruner | 94.2% | 95.2% | **96.0%** |
| HoloV | 93.1% | 94.8% | **96.5%** |

*Table 7.* Comparison with the position of merged tokens.

| Position Type | Average |
|---|---|
| First | 96.3% |
| Median | **96.5%** |
| Mean | 96.0% |
| Last | 95.2% |

*Table 8.* Computational overhead of distinctive anchor token selection.

| $n_{vis}$ | Overhead Ratio | Anchor Selection (FLOPs) | MLLMs Calculation (FLOPs) |
|---|---|---|---|
| 192 | 0.074% | | 1.839 T |
| 128 | 0.096% | 1.359 G | 1.417 T |
| 64 | 0.136% | | 0.997 T |

*Table 9.* End-to-end latency analysis on the GQA (Hudson & Manning, 2019) dataset.

| Method | Total Inference | | Prefill | | Accuracy | |
|---|---|---|---|---|---|---|
| | Time | Speedup | Time | Speedup | Acc. | Δ |
| LLaVA-1.5-7B | 6m05s | 1.00× | 227.8ms | 1.00× | 61.9 | - |
| HoloV (192 tokens) | 4m23s | 1.39× | 178.2ms | 1.28× | 58.6 | -5.33% |
| + ours | 4m28s | 1.36× | 181.4ms | 1.26× | 61.0 | -1.45% |
| HoloV (128 tokens) | 3m58s | 1.53× | 158.2ms | 1.44× | 57.5 | -7.11% |
| + ours | 4m02s | 1.51× | 164.0ms | 1.39× | 60.8 | -1.78% |
| HoloV (64 tokens) | 3m39s | 1.67× | 145.6ms | 1.56× | 55.1 | -10.99% |
| + ours | 3m43s | 1.64× | 151.1ms | 1.51× | 59.0 | -4.68% |

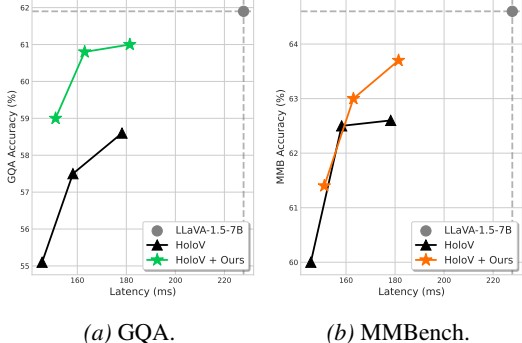

*(a)* GQA.      *(b)* MMBench.

*Figure 4.* Accuracy-latency trade-off curves on (a) GQA (Hudson & Manning, 2019) and (b) MMBench (Liu et al., 2024b) by varying the number of retained visual tokens.

gible compared to the total computational cost of MLLMs. As shown in Table 8, the anchor selection requires a constant 1.359 GFLOPs for LLaVA-1.5-7B (Liu et al., 2024a). Even with the most aggressive reduction to 64 tokens, it constitutes only 0.136% of the total inference FLOPs. The attention calibration is also computationally efficient in that the calibration requires an $N \times N$ matrix. The matrix is calculated once prior to the LLM input stage, and then applied to each layer, incurring minimal additional cost. We also show in Fig. 4 the trade-off between accuracy and latency. The results demonstrate that our method achieves a superior trade-off compared to the baseline. While incurring only a marginal increase in latency, it achieves significant accuracy gains. To further support our efficiency, we measure the end-to-end latency, including both the total inference time and the prefill time, on the GQA dataset using an $8\times$ NVIDIA A5000 GPU setup. As shown in Table 9, integrating our framework introduces only a marginal latency overhead over the baseline HoloV. For instance, at a retention of 64 tokens, it adds merely 4 seconds to the total inference time and 5.5ms to the prefill time. Despite this negligible cost, our framework narrows the accuracy gap to the full-token model, reducing the degradation of HoloV from 10.99% to 4.68% at 64 tokens. These results support that our method preserves the practical speedup of VTR while delivering significant accuracy gains.

## 5. Conclusion

In this paper, we have proposed RESTORE, a novel framework to improve existing VTR methods for efficient MLLM inference. We have identified a critical yet overlooked issue in previous methods, attentional distortion caused by position assignment, and addressed it through an attention calibration mechanism. We have also introduced a distinctive anchor selection for token merging to mitigate information loss during token merging. Extensive experiments demonstrate that integrating our framework with various baselines consistently yields state-of-the-art performance across multiple benchmarks, maintaining efficiency.

## Impact Statement

Our method focuses on improving the computational efficiency of MLLMs. This direction has the potential to reduce energy consumption during inference and facilitate deployment in resource-constrained environments (*e.g.,* mobile phones). We do not predict any negative societal consequences or ethical issues specific to this work.

## Acknowledgement

This work was partly supported by Institute of Information & Communications Technology Planning & Evaluation (IITP) grant funded by the Korea government (MSIT) (No.RS-2022-00143524, Development of Fundamental Technology and Integrated Solution for Next-Generation Automatic Artificial Intelligence System, No.RS-2025-09942968, AI Semiconductor Innovation Lab (Yonsei University)), the National Research Foundation of Korea (NRF) grant funded by the Korea government (MSIT) (RS-2025-02216328), and the KIST Institutional Program (Project No.2E33001-24-086).

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

# Appendix

## A. Attention calibration for Qwen2.5-VL with M-RoPE

In standard 1D RoPE, our decay function $\mathcal{D}$ (Eq. 3) averages cosine terms across all frequency dimensions. In M-RoPE, the half-head dimension $d_h/2$ is partitioned into $G$ independent axes (e.g., temporal, height, width). Each axis $g \in \{1, \ldots, G\}$ is allocated $d_g$ dimensions and has its own positional distance $\Delta_g$. To extend our attention calibration to M-RoPE, we generalize the standard 1D RoPE formulation into a weighted sum of per-axis decay functions:

$$\mathcal{D}_{M\text{-}RoPE} = \sum_{g=1}^{G} w_g \mathcal{D}_g(\Delta_g), \tag{9}$$

where $\mathcal{D}_g(\Delta_g) = \frac{1}{|\Omega_g|} \sum_{j \in \Omega_g} \cos(\Delta_g \cdot \theta_j)$ computes the specific decay for axis $g$. Here, $\Omega_g$ represents the set of frequency indices assigned to axis $g$ ($|\Omega_g| = d_g$), and $\theta_j$ is the rotation frequency for index $j$. The weight $w_g = \frac{2d_g}{d_h}$ denotes the proportion of dimensions allocated to axis $g$, which guarantees $\sum_{g=1}^{G} w_g = 1$ and preserves the scale of the standard 1D RoPE. This formulation preserves our fundamental principle: attention inherently decays as positional distance increases, requiring a proportional calibration bias in the VTR scenario.

## B. Theoretical Analysis of Textual Attention Amplification

In this section, we provide a theoretical analysis to elucidate the disproportionate amplification of textual attention described in Fig. 1(d). Specifically, we mathematically demonstrate how pruning visual tokens inadvertently exacerbates the initial attention disparity caused by positional bias, leading the model to over-attend to text tokens.

Let $S$ be the original set of all tokens ($|S| = N_{sys} + N_{vis} + N_{txt}$) and $P$ be the set of pruned visual tokens. For any token $i$, let $z_i$ be its attention logit. The original attention weight is defined as:

$$a_i = \frac{\exp(z_i)}{Z}, \quad \text{where} \quad Z = \sum_{j \in S} \exp(z_j). \tag{10}$$

After pruning the visual tokens from $N_{vis}$ to $n_{vis}$, the partition function reduces to $Z' = Z - \sum_{k \in P} \exp(z_k)$. The new attention weight for a remaining token $i \in S \setminus P$ is given by:

$$a_i' = \frac{\exp(z_i)}{Z'}. \tag{11}$$

Consequently, the amplification in the attention weight for token $i$ can be expressed as:

$$\Delta a_i = a_i' - a_i = \exp(z_i) \left( \frac{1}{Z'} - \frac{1}{Z} \right) = a_i \left( \frac{Z - Z'}{Z'} \right). \tag{12}$$

Since $Z > Z'$, the term $\frac{Z-Z'}{Z'}$ is a positive constant across all remaining tokens. This derivation demonstrates that the redistributed probability mass from the pruned tokens is not shared uniformly. Instead, the absolute increase $\Delta a_i$ is directly proportional to its original weight $a_i$. That is, tokens that originally possessed larger attention weights absorb a larger portion of the redistributed weights.

## C. More Results

**Text-aware baselines.** In the main paper, we integrate our framework with text-agnostic VTR methods, as they avoid the computational overhead of computing attention weights within the LLM layers and are thus more efficient. Nevertheless, our framework is not inherently restricted to text-agnostic methods. It is equally feasible for text-aware approaches. This is because our attention calibration addresses the positional and attentional distortions that arise in any reduced token sequence, irrespective of how the retained tokens are selected, while our distinctive anchor selection operates on the merging stage that is orthogonal to the underlying pruning criterion. To verify this, we integrate our framework with three representative text-aware methods, including FastV (Chen et al., 2024), PDrop (Xing et al., 2024), and SparseVLM (Zhang et al., 2025b).

*Table 10.* Comparison with text-aware VTR methods for LLaVA-1.5-7B (Liu et al., 2024a) on eight benchmarks and the average score across them. For baselines, all experiment results are re-implemented from their official codebases under the same environments. Best results at each reduction ratio in Bold.

| Method | Average | GQA | MMB | MME | POPE | SQA$^{\text{IMG}}$ | VQA$^{\text{V2}}$ | VQA$^{\text{Text}}$ | SEED |
|---|---|---|---|---|---|---|---|---|---|
| *Using All Visual Tokens, 576 Tokens (100%)* | | | | | | | | | |
| LLaVA-1.5-7B | 100% | 61.9 | 64.6 | 1862 | 85.9 | 69.5 | 78.5 | 58.2 | 58.6 |
| *Retain 192 Tokens (33.3%)* | | | | | | | | | |
| FastV (ECCV24) | 96.0% | 57.1 | 64.4 | 1821 | 75.8 | 68.9 | 74.7 | 57.8 | 56.3 |
| + ours | 99.0% | 60.8 | 64.0 | 1853 | 86.0 | 68.4 | 77.4 | 57.4 | 58.1 |
| PDrop (CVPR25) | 96.8% | 58.0 | 62.9 | 1790 | 84.0 | 68.9 | 76.0 | 57.1 | 55.8 |
| + ours | 99.1% | 61.0 | 64.1 | 1829 | 86.1 | 68.8 | 78.0 | 57.4 | 58.4 |
| SparseVLM (ICML25) | 98.1% | 59.5 | 64.2 | 1782 | 85.4 | 68.7 | 77.0 | 57.7 | 57.3 |
| + ours | **99.4%** | 61.4 | 64.1 | 1831 | 86.5 | 69.1 | 78.1 | 57.6 | 58.5 |
| *Retain 128 Tokens (22.2%)* | | | | | | | | | |
| FastV (ECCV24) | 91.8% | 54.1 | 63.1 | 1694 | 68.3 | 69.3 | 70.7 | 56.3 | 54.0 |
| + ours | 97.9% | 59.7 | 63.6 | 1818 | 85.4 | 68.3 | 76.3 | 56.9 | 57.5 |
| PDrop (CVPR25) | 84.5% | 51.6 | 54.9 | 1403 | 67.8 | 68.8 | 65.6 | 52.6 | 47.0 |
| + ours | 95.3% | 57.8 | 63.3 | 1721 | 84.2 | 68.0 | 74.6 | 53.2 | 56.4 |
| SparseVLM (ICML25) | 97.2% | 58.4 | 64.4 | 1761 | 85.0 | 68.7 | 76.2 | 56.6 | 56.9 |
| + ours | **98.5%** | 60.9 | 63.0 | 1806 | 86.7 | 68.5 | 77.7 | 56.3 | 58.4 |
| *Retain 64 Tokens (11.1%)* | | | | | | | | | |
| FastV (ECCV24) | 75.2% | 46.4 | 51.4 | 1284 | 36.1 | 69.9 | 56.2 | 51.6 | 43.9 |
| + ours | 89.9% | 54.1 | 61.2 | 1621 | 75.9 | 66.4 | 69.0 | 51.5 | 52.8 |
| PDrop (CVPR25) | 80.9% | 49.0 | 55.6 | 1404 | 55.5 | 69.5 | 60.1 | 50.6 | 46.1 |
| + ours | 91.7% | 54.9 | 61.6 | 1674 | 78.2 | 67.6 | 71.1 | 52.3 | 53.5 |
| SparseVLM (ICML25) | 90.6% | 53.8 | 60.2 | 1591 | 77.5 | 69.7 | 70.3 | 53.5 | 51.2 |
| + ours | **94.5%** | 57.1 | 61.9 | 1721 | 83.4 | 68.6 | 73.6 | 53.4 | 55.3 |

As shown in Table 10, our method consistently improves the accuracy of all text-aware baselines across every reduction ratio. The improvement becomes more pronounced under aggressive reduction. For instance, at a retention of 64 tokens, our framework improves FastV by 14.7% and PDrop by 10.8%. These results demonstrate that our framework is broadly compatible with diverse VTR strategies, rectifying the distortions overlooked by both text-aware and text-agnostic methods.

**Experiment on fine-grained visual perception task.** To examine whether our framework remains effective on tasks that demand fine-grained visual perception, we evaluate it on the OCRBench (Liu et al., 2024c) dataset, which requires precise recognition of text within images. Unlike coarse-grained recognition, OCR is highly sensitive to the loss of local visual details, making it a challenging testbed for VTR methods. We integrate our framework with four text-agnostic baselines, VisionZip, DivPrune, VisPruner, and HoloV, under three retention ratios ($n_{vis} = 192, 128$, and $64$). As shown in Table 11, our framework consistently improves the OCRBench score across nearly all baselines and retention ratios. The gains are substantial for several baselines; for instance, our framework improves DivPrune by 24 points at $n_{vis} = 128$ and VisionZip by 11 points at $n_{vis} = 64$. The only minor exception arises for VisPruner under the most aggressive setting ($n_{vis} = 64$), where the extreme compression leaves too few tokens to preserve the fine-grained textual cues. Overall, these results indicate that rectifying positional and attentional distortions helps the model retain fine-grained visual information, demonstrating that our framework is effective even on perception-intensive tasks.

*Table 11.* Comparison with VTR methods on the OCRBench (Liu et al., 2024c) dataset for LLaVA-1.5-7B (Liu et al., 2024a), evaluating fine-grained visual perception. Best results at each retention ratio in Bold.

| Retained Tokens | Method | VisionZip | DivPrune | VisPruner | HoloV |
|---|---|---|---|---|---|
| $n_{vis} = 192$ | Vanilla | 286 | 281 | 295 | **295** |
| | + ours | **290** | **293** | **298** | **295** |
| $n_{vis} = 128$ | Vanilla | 285 | 264 | 290 | 288 |
| | + ours | **287** | **288** | **294** | **301** |
| $n_{vis} = 64$ | Vanilla | 258 | 257 | **287** | 279 |
| | + ours | **269** | **267** | 273 | **283** |

*Table 12.* Ablation on the pruning ratio $\gamma$ on TextVQA (Singh et al., 2019) at $n_{vis} = 192$ for LLaVA-1.5-7B (Liu et al., 2024a). $\gamma = 0$ corresponds to merging only and $\gamma = 1$ to pruning only.

| Method | Baseline | $\gamma = 0$ | $\gamma = 0.25$ | $\gamma = 0.5$ | $\gamma = 0.75$ | $\gamma = 1$ |
|---|---|---|---|---|---|---|
| VisionZip | 55.8 | 53.6 | 54.6 | 54.9 | 54.7 | 55.8 |
| DivPrune | 55.7 | 56.8 | 56.5 | 56.5 | 56.5 | 56.8 |
| VisPruner | 57.7 | 56.8 | 56.9 | 57.0 | 57.0 | 57.7 |
| HoloV | 55.8 | 56.8 | 56.9 | 57.2 | 57.3 | 57.2 |

*Table 13.* Compatibility of our framework with the layer-wise, adaptive pruning method FlowCut (Tong et al., 2025) on LLaVA-1.5-7B (Liu et al., 2024a). We report results at two retention ratios with three pruning ratios $\gamma$. Best results at each retention ratio in Bold.

| Method | Average | GQA | MMB | MME | POPE | SQA$^{\text{IMG}}$ | VQA$^{\text{V2}}$ | VQA$^{\text{Text}}$ | SEED |
|---|---|---|---|---|---|---|---|---|---|
| *Using All Visual Tokens, 576 Tokens* **(100%)** | | | | | | | | | |
| LLaVA-1.5-7B | 100% | 61.9 | 64.6 | 1862 | 85.9 | 69.5 | 78.5 | 58.2 | 58.6 |
| *Retain 128 Tokens* **(22.2%)** | | | | | | | | | |
| FlowCut | 97.0% | 58.5 | 62.1 | 1792 | 85.2 | 68.6 | 76.0 | 57.3 | 56.2 |
| + ours ($\gamma = 0.5$) | 97.2% | 60.6 | 63.3 | 1736 | 86.3 | 66.9 | 77.3 | 55.7 | 57.3 |
| + ours ($\gamma = 0.75$) | **97.7%** | 60.4 | 63.2 | 1764 | 86.3 | 68.3 | 77.4 | 55.8 | 57.5 |
| + ours ($\gamma = 0.9$) | 97.5% | 59.8 | 62.5 | 1819 | 85.3 | 68.3 | 77.2 | 55.6 | 57.3 |
| *Retain 64 Tokens* **(11.1%)** | | | | | | | | | |
| FlowCut | 93.7% | 55.6 | 60.8 | 1744 | 80.2 | 69.1 | 72.8 | 55.6 | 53.5 |
| + ours ($\gamma = 0.5$) | 95.3% | 58.9 | 62.2 | 1690 | 84.8 | 67.9 | 75.6 | 53.3 | 56.2 |
| + ours ($\gamma = 0.75$) | **95.7%** | 58.7 | 63.1 | 1749 | 83.8 | 68.0 | 75.5 | 53.9 | 55.5 |
| + ours ($\gamma = 0.9$) | 94.2% | 56.9 | 61.4 | 1746 | 81.9 | 67.8 | 74.3 | 54.0 | 54.4 |

**Analysis of performance regression on TextVQA.** While our framework consistently improves the overall average accuracy, we observe that it occasionally underperforms the baseline on TextVQA (Singh et al., 2019): for example, when integrated with VisionZip at a retention of 192 tokens (Table 1) and with LLaVA-NeXT-7B at a retention of 160 tokens (Table 2). We investigate the cause of these regressions and find that they do not originate from the attention calibration over-suppressing text-relevant tokens. Instead, they are attributed to the feature averaging in the token merging stage. TextVQA requires reading fine-grained, highly localized text within images, and such information is typically concentrated in a small number of visual tokens. When these tokens are merged, averaging their features with surrounding tokens dilutes the localized high-frequency details that are essential for accurate text recognition.

To verify this, we conduct an ablation on the pruning ratio $\gamma$ while keeping the attention calibration enabled, evaluated on TextVQA at $n_{vis} = 192$. Since $\gamma$ governs the token budget allocated to pruning, a larger $\gamma$ retains more tokens intact and merges fewer of them. As shown in Table 12, increasing $\gamma$ consistently alleviates the regression, and bypassing merging entirely ($\gamma = 1$) recovers the performance to, or above, the baseline across all methods. As the attention calibration remains fixed throughout this ablation, the result confirms that the regression stems from the merging stage rather than from the calibration. We therefore note that, for tasks demanding fine-grained visual perception, the hybrid reduction strategy would benefit from adaptively suppressing merging to preserve critical localized information, which we leave as a promising direction for future work.

**Compatibility with layer-wise and adaptive pruning methods such as FlowCut.** Our framework is readily compatible with layer-wise, adaptive pruning methods such as FlowCut (Tong et al., 2025), which estimate token importance using multiple signals across layers. Although FlowCut aggregates multi-faceted criteria, the resulting importance ultimately reduces to a single scalar score per token at a given layer. Therefore, at any intermediate pruning stage $i$ that retains $n_{vis,i}$ tokens, we can directly apply our hybrid reduction strategy. Specifically, we retain the top $\gamma \cdot n_{vis,i}$ tokens by pruning based on the FlowCut scores, and construct the remaining $(1 - \gamma) \cdot n_{vis,i}$ tokens by merging with our distinctive anchor selection, followed by attention calibration.

Table 13 reports the results of FlowCut and its integration with our framework on eight benchmarks at two retention ratios ($n_{vis} = 128$ and 64). Since FlowCut performs pruning over multiple stages, iterative merging may accumulate feature distortion; we therefore additionally examine higher pruning ratios ($\gamma = 0.75$ and 0.9) that allocate a larger token budget to pruning. Our framework consistently improves the performance of FlowCut across all settings. Notably, $\gamma = 0.75$ achieves the best overall accuracy, indicating that a moderately pruning-oriented allocation is favorable for multi-stage adaptive

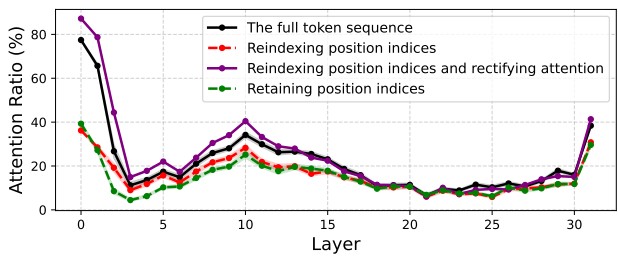 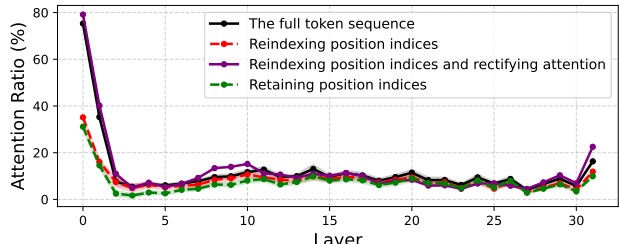

*(a)* Visual-to-visual attention.                         *(b)* Text-to-visual attention.

*Figure 5.* Comparison of average attention proportions assigned to visual tokens within the LLM (a) when the query is a visual token and (b) when the query is a text token. We add a case of reindexing position indices with our proposed calibration.

*Table 14.* Ablation study on the components of our framework with detailed benchmark results (expanding Table 4).

| | Incorporating Merging | Retaining Position | Attention Calibration | Average | GQA | MMB | MME | POPE | SQA$^{\text{IMG}}$ | VQA$^{\text{V2}}$ | VQA$^{\text{Text}}$ | SEED |
|---|---|---|---|---|---|---|---|---|---|---|---|---|
| ① | ✗ | ✗ | ✗ | 92.2% | 55.1 | 60.0 | 1699 | 76.8 | 68.6 | 72.3 | 54.9 | 52.5 |
| ② | ✗ | ✗ | ✓ | 91.0% | 54.5 | 57.6 | 1688 | 78.6 | 68.0 | 71.8 | 52.2 | 52.4 |
| ③ | ✗ | ✓ | ✗ | 90.6% | 54.5 | 60.2 | 1539 | 76.6 | 67.0 | 71.6 | 54.5 | 53.2 |
| ④ | ✗ | ✓ | ✓ | 93.1% | 55.5 | 62.1 | 1739 | 76.0 | 68.5 | 72.3 | 54.8 | 53.9 |
| ⑤ | ✓ | ✗ | ✗ | 93.5% | 56.6 | 59.6 | 1679 | 82.1 | 68.7 | 73.2 | 55.4 | 53.4 |
| ⑥ | ✓ | ✗ | ✓ | 92.9% | 57.1 | 62.1 | 1530 | 87.6 | 66.6 | 73.4 | 53.0 | 53.0 |
| ⑦ | ✓ | ✓ | ✗ | 90.1% | 54.2 | 59.0 | 1419 | 83.4 | 66.4 | 71.0 | 53.4 | 53.4 |
| ⑧ | ✓ | ✓ | ✓ | **96.5%** | 59.0 | 61.9 | 1787 | 84.9 | 68.0 | 75.6 | 55.4 | 56.7 |

methods, as it limits the distortion introduced by repeated merging while still benefiting from global context aggregation. These results demonstrate that our calibration and distinctive anchor selection generalize beyond single-stage reduction to layer-wise, adaptive pruning methods.

**Detailed results of tables in main text.** We provide detailed results of Tables 4-7 in Tables 14-17, respectively.

## D. More Discussions

**Attention calibration with reindexing.** As observed in Fig. 2, the reindexing strategy yields an attention distribution across layers that is closer to the full token sequence compared to retaining position indices. This raises a natural question: can applying attention calibration to the reindexing strategy further improve alignment? We present the results of this experiment in Fig. 5. Contrary to the beneficial effect observed with retaining indices, applying attention calibration to reindexing causes the attention weights assigned to visual tokens to significantly overshoot those of the full token sequence (solid purple line). This leads to an excessive dominance of visual context, where visual tokens overshadow the necessary textual information. This imbalance leads to performance degradation, as evidenced by the quantitative results in Row ② of Table 4. The accuracy drops to 91.0%, falling behind the baseline of 92.2% (Row ①). This finding supports that our calibration term is specifically designed to counteract the positional decay inherent to retaining original indices.

**Quantitative analysis of information loss in token merging.** To validate that our distinctive anchor selection mitigates information loss during merging, we provide a quantitative analysis based on the Hausdorff distance (Huttenlocher et al., 2002; Alvar et al., 2025), which measures how well the merged token set preserves the original visual information. The standard Hausdorff distance computes the distance from each original token to its nearest anchor, but treats all anchors equally and thus disregards the broader semantic coverage of large merged groups. Since a merged anchor $y_j$ serves as a proxy for a cluster of $s_j$ tokens, a larger cluster inherently spans a wider semantic region, and the distance penalty to $y_j$ should be discounted in proportion to its cluster size. To reflect this, we define a weighted Hausdorff distance between the set of original visual tokens $X$ and the set of merged anchor tokens $Y$ as:

$$h_w(X, Y) = \max_{x \in X} \min_{y_j \in Y} \frac{1 - \cos(x, y_j)}{s_j}, \tag{13}$$

*Table 15.* Full comparison results with attention calibration strategies (expanding Table 5). 'w/o' denotes no calibration, '$s_n$' denotes proportional calibration, and 'Ours' denotes our proposed strategy $s_n(c - \mathcal{D}(|m - n|))$.

| Method | Calibration | Average | GQA | MMB | MME | POPE | SQA$^{IMG}$ | VQA$^{V2}$ | VQA$^{Text}$ | SEED |
|---|---|---|---|---|---|---|---|---|---|---|
| | w/o | 89.5% | 54.0 | 58.3 | 1444 | 82.9 | 66.2 | 70.3 | 52.1 | 52.6 |
| VisionZip | $s_n$ | 94.4% | 58.3 | 61.4 | 1660 | 83.8 | 68.2 | 74.4 | 53.5 | 55.2 |
| | Ours | **94.8%** | 58.5 | 62.3 | 1726 | 84.6 | 68.0 | 74.3 | 52.2 | 55.2 |
| | w/o | 88.9% | 53.2 | 56.8 | 1400 | 84.8 | 64.8 | 70.5 | 51.9 | 53.3 |
| DivPrune | $s_n$ | 94.9% | 58.5 | 62.1 | 1676 | 83.8 | 67.0 | 74.6 | 54.6 | 56.1 |
| | Ours | **95.9%** | 59.0 | 62.2 | 1748 | 84.8 | 68.0 | 75.0 | 54.4 | 56.3 |
| | w/o | 88.7% | 53.2 | 56.6 | 1361 | 84.3 | 65.2 | 70.2 | 53.0 | 53.2 |
| VisPruner | $s_n$ | 95.1% | 58.9 | 60.7 | 1661 | 84.2 | 67.8 | 75.1 | 55.6 | 56.2 |
| | Ours | **96.0%** | 59.2 | 61.6 | 1722 | 85.1 | 68.0 | 75.4 | 55.4 | 56.5 |
| | w/o | 90.1% | 54.2 | 59.0 | 1419 | 83.4 | 66.4 | 71.0 | 53.4 | 53.4 |
| HoloV | $s_n$ | 95.3% | 58.9 | 61.1 | 1694 | 83.9 | 67.3 | 75.2 | 55.5 | 56.1 |
| | Ours | **96.5%** | 59.0 | 61.9 | 1787 | 84.9 | 68.0 | 75.6 | 55.4 | 56.7 |

*Table 16.* Full comparison results with reduction types (Expanding Table 6). P denotes Pruning and M denotes Merging. 'unif.' stands for uniform anchor sampling, and 'dist.' stands for distinctive anchor sampling (Ours).

| Method | Reduction Type | Average | GQA | MMB | MME | POPE | SQA$^{IMG}$ | VQA$^{V2}$ | VQA$^{Text}$ | SEED |
|---|---|---|---|---|---|---|---|---|---|---|
| | P | - | - | - | - | - | - | - | - | - |
| VisionZip | P + M (unif.) | 94.4% | 58.2 | 62.3 | 1663 | 83.0 | 68.8 | 74.6 | 52.7 | 55.3 |
| | P + M (dist.) | **94.8%** | 58.5 | 62.3 | 1726 | 84.6 | 68.0 | 74.3 | 52.2 | 55.2 |
| | P | 95.8% | 59.0 | 60.8 | 1778 | 84.1 | 69.3 | 74.6 | 54.6 | 55.6 |
| DivPrune | P + M (unif.) | 94.9% | 57.9 | 62.6 | 1699 | 83.7 | 68.2 | 74.1 | 53.4 | 56.0 |
| | P + M (dist.) | **95.9%** | 59.0 | 62.2 | 1748 | 84.8 | 68.0 | 75.0 | 54.4 | 56.3 |
| | P | 94.2% | 57.0 | 61.1 | 1760 | 80.0 | 69.4 | 73.1 | 54.9 | 54.0 |
| VisPruner | P + M (unif.) | 95.2% | 58.1 | 62.6 | 1721 | 83.2 | 68.6 | 74.5 | 54.0 | 55.7 |
| | P + M (dist.) | **96.0%** | 59.2 | 61.6 | 1722 | 85.1 | 68.0 | 75.4 | 55.4 | 56.5 |
| | P | 93.1% | 55.5 | 62.1 | 1739 | 76.0 | 68.5 | 72.3 | 54.8 | 53.9 |
| HoloV | P + M (unif.) | 94.8% | 57.4 | 62.8 | 1690 | 82.2 | 68.7 | 74.5 | 54.2 | 55.9 |
| | P + M (dist.) | **96.5%** | 59.0 | 61.9 | 1787 | 84.9 | 68.0 | 75.6 | 55.4 | 56.7 |

*Table 17.* Full comparison results with the position of tokens (Expanding Table 7).

| Position Type | Average | GQA | MMB | MME | POPE | SQA$^{IMG}$ | VQA$^{V2}$ | VQA$^{Text}$ | SEED |
|---|---|---|---|---|---|---|---|---|---|
| First | 96.3% | 58.6 | 61.9 | 1732 | 85.6 | 69.0 | 75.4 | 55.8 | 56.1 |
| Median | **96.5%** | 59.0 | 61.9 | 1787 | 84.9 | 68.0 | 75.6 | 55.4 | 56.7 |
| Mean | 96.0% | 58.2 | 62.2 | 1741 | 85.3 | 68.1 | 75.6 | 55.0 | 56.3 |
| Last | 95.2% | 57.7 | 62.0 | 1695 | 85.1 | 68.0 | 74.5 | 55.3 | 55.6 |

where $\cos(\cdot, \cdot)$ denotes the cosine similarity and $s_j$ is the number of tokens merged into the anchor $y_j$. A lower $h_w$ indicates that the merged tokens better cover the original visual information, implying less information loss.

We measure $h_w$ on four baselines under three configurations: pruning only, pruning combined with merging using uniformly sampled anchors, and pruning combined with our distinctive anchors. The values are averaged over 100 non-overlapping images from the GQA (Hudson & Manning, 2019) dataset using LLaVA-1.5-7B (Liu et al., 2024a). As shown in Table 18, pruning alone incurs the highest information loss, since discarding tokens entirely removes their information. While uniform merging substantially reduces this loss by aggregating tokens, our distinctive anchor selection achieves the lowest $h_w$ across all baselines. This confirms that selecting representative and discriminative anchors minimizes information loss during feature averaging, complementing the attentional and positional rectification of our framework.

**Hyperparameter $\gamma$.** The pruning ratio $\gamma$ determines the allocation of the token budget between pruning and merging stages in our hybrid reduction strategy. We investigate the impact of varying $\gamma$ on performance (Table 19). The results indicate that a balanced allocation ($\gamma = 0.5$) yields optimal performance, demonstrating that both pruning and merging

*Table 18.* Quantitative analysis of information loss in token merging, measured by the weighted Hausdorff distance $h_w$ (Eq. 13) on LLaVA-1.5-7B (Liu et al., 2024a). Lower is better. Values are averaged over 100 images from the GQA (Hudson & Manning, 2019) dataset. Best results in Bold.

| Baseline | Pruning Only | P + M (Uniform) | P + M (Distinctive) |
|---|---|---|---|
| VisionZip | 0.7168 | 0.0169 | **0.0077** |
| DivPrune | 0.2889 | 0.0177 | **0.0078** |
| VisPruner | 0.4590 | 0.0169 | **0.0078** |
| HoloV | 0.6701 | 0.0167 | **0.0076** |

contribute significantly to effective token reduction. We conjecture that this is because the token pruning preserves intact information of selected tokens while the token merging captures global context via aggregating multiple tokens. Extreme allocations, such as $\gamma = 0.0$ (merging only) or $\gamma = 1.0$ (pruning only), lead to suboptimal performance, highlighting the importance of a hybrid approach.

*Table 19.* Ablation study on the hyperparameter $\gamma$.

| | Average | GQA | MMB | MME | POPE | SQA$^{\text{IMG}}$ | VQA$^{\text{V2}}$ | VQA$^{\text{Text}}$ | SEED |
|---|---|---|---|---|---|---|---|---|---|
| LLaVA-1.5-7B | 100% | 61.9 | 64.6 | 1862 | 85.9 | 69.5 | 78.5 | 58.2 | 58.6 |
| HoloV (Baseline) | 92.2% | 55.1 | 60.0 | 1699 | 76.8 | 68.6 | 72.3 | 54.9 | 52.5 |
| $\gamma = 0$ | 95.7% | 59.4 | 62.8 | 1727 | 84.4 | 68.3 | 74.6 | 53.4 | 56.7 |
| $\gamma = 0.25$ | 96.2% | 59.4 | 62.8 | 1703 | 84.7 | 69.3 | 75.7 | 54.9 | 56.4 |
| $\gamma = 0.5$ | **96.5%** | 59.0 | 61.9 | 1787 | 84.9 | 68.0 | 75.6 | 55.4 | 56.7 |
| $\gamma = 0.75$ | 95.4% | 57.3 | 63.7 | 1740 | 82.8 | 68.1 | 74.6 | 55.1 | 55.5 |
| $\gamma = 1.0$ | 93.1% | 55.5 | 62.1 | 1739 | 76.0 | 68.5 | 72.3 | 54.8 | 53.9 |

# E. Detailed Computational Complexity Analysis

We provide a detailed calculation of the FLOPs presented in Table 8 and expanded in Table 20. To precisely estimate the inference cost, we compare the FLOPs occurring in the visual encoder, anchor selection, and the LLM. We exclude the projector from the total FLOPs calculation as its computational cost is negligible compared to the LLM and visual encoder.

*Table 20.* Detailed computational complexity analysis. The Total MLLM FLOPs is the sum of the Visual Encoder (constant) and LLM Decoder (variable based on $n_{vis}$). The projector's cost is omitted as it is negligible. $N_{sys} = 35$, $N_{vis} = 576$, $N_{txt} = 25$.

| Module | Configuration / Formula | Params | Target Visual Tokens ($n_{vis}$) | | |
|---|---|---|---|---|---|
| | | | **192** | **128** | **64** |
| **Visual Encoder** (ViT-L/14, 24 Layers) | Input Tokens ($N_{vis}$) 
 Hidden Size ($D$) / FFN ($F$) 
 FLOPs (Attn) $\approx 4N_{vis}D^2 + 2N_{vis}^2 D$ 
 FLOPs (MLP) $\approx 2N_{vis}FD$ | 576 
 1024 / 4096 
 - 
 - | 576 (Fixed) 
 Constant Params 

 Constant Cost | | |
| | **FLOPs** | - | **0.190 TFLOPs** | | |
| **Projector** | Linear Projection | - | *Negligible* | | |
| **LLM Decoder** (LLaMA-7B, 32 Layers) | System + Text ($N_{sys} + N_{txt}$) 
 **Total Input Tokens** ($N$) 
 Hidden Size ($D$) / FFN ($F$) 
 FLOPs (Attn) $\approx 4ND^2 + 2N^2D$ 
 FLOPs (MLP) $\approx 3NFD$ | 60 
 $60 + n_{vis}$ 
 4096 / 11008 
 - 
 - | 60 (Fixed) 
 **252**    **188**    **124** 
 Constant Params 

 Variable Cost | | |
| | **FLOPs** | - | 1.649 TFLOPs | 1.227 TFLOPs | 0.807 TFLOPs |
| **Total MLLM** | Visual Encoder + LLM | - | **1.839 T (TFLOPs)** | **1.417 T (TFLOPs)** | **0.997 T (TFLOPs)** |
| **Overhead** | Anchor Selection Cost | - | 1.359 GFLOPs ($\approx 0.001$ TFLOPs) | | |
| | **Ratio** (Overhead / Total) | - | **0.074%** | **0.096%** | **0.136%** |