# OpenReview forum: "Improving Visual Token Reduction via Rectifying Distortions for Efficient Multimodal LLM Inference"
_ICML.cc/2026/Conference — ICML 2026 regular_

### Official Review · Reviewer_BtE8 · 2026-02-20

**Soundness:** 2
**Presentation:** 2
**Significance:** 3
**Originality:** 2
**Overall Recommendation:** 5
**Confidence:** 3

**Summary:**

This paper identifies attentional and positional distortions in visual token reduction (VTR) for MLLMs, where existing methods fail to preserve total attention weights to visual tokens. The authors propose a distance-aware attention calibration method to counteract RoPE decay and a distinctive anchor selection strategy for token merging. Experiments across multiple benchmarks demonstrate consistent improvements over existing VTR baselines.

**Compliance With Llm Reviewing Policy:**

Affirmed.

**Final Justification:**

The author's reply has basically resolved my issue, and I will maintain a positive score.

**Key Questions For Authors:**

1. You set `c = 2` to ensure the calibration term remains positive. Have you explored whether `c` should vary with model dimension, head count, or sequence length? The decay function D(|m-n|) oscillates (Figure 3); does `c = 2` remain optimal when |m-n| exceeds 2π/θ_j for the smallest frequency θ_j?

2. Table 6 shows Median outperforms Last by 1.3% average accuracy. Yet your attention calibration in Eq. 4 uses relative distance |m-n|, meaning a Median-assigned anchor at position 50 receives different calibration than a Last-assigned anchor at position 100. Is the Median advantage due to better spatial representation, or simply because Median positions happen to fall in a calibration "sweet spot" where (c - D(|m-n|)) is larger?

3. Your experiments use N_vis = 576 (ViT-L/14 with 336px images). Modern MLLMs often process 1024×1024 images with N_vis = 1024 or higher. At these scales, the O(N_vis^2) correlation computation for anchor selection (Eq. 5-8) becomes expensive. Have you measured wall-clock latency overhead (not just FLOPs) for N_vis > 1000? Does the distinctive anchor selection remain tractable?

4. Your framework integrates with text-agnostic methods only, arguing they are "more efficient." However, text-aware methods like SparseVLM (ICML 2025) achieve strong results (Table 1: 98.1% at 192 tokens vs your 98.8% with HoloV+Ours). The gap is small. Have you attempted applying your calibration to text-aware baselines? The text query could provide additional signal for the distance-aware term.

5. Your calibration boosts distant visual tokens to compensate for RoPE decay. Could this cause over-attention to irrelevant background regions? Figure 4 shows improved accuracy, but does the model ever attend *too much* to peripheral visual tokens at the expense of central objects? A failure case analysis would illuminate the method's boundaries.

**Limitations:**

see weaknesses.

**Strengths And Weaknesses:**

**Strengths:**
- Analysis of softmax redistribution and RoPE positional bias causing attention attenuation is compelling with clear empirical evidence in Figure 2; Figure 2 demonstrates that neither reindexing nor retaining original indices preserves baseline attention levels
- Calibration term derived from first principles modeling RoPE decay function; derivation from RoPE's rotation components provides principled foundation
- Distinctive anchor selection formulation elegantly combines representativeness with discriminativeness; plug-and-play compatibility with existing VTR methods enhances practical value
- Ablation studies in Table 4 are thorough, showing position retention and attention calibration are complementary components; framework demonstrates practical utility with 4.3% improvement at aggressive reduction (64 tokens)
- Figure 1 effectively visualizes the distortion problem; mathematical notation is consistent and precise throughout
- Position-aware calibration with redundancy-aware merging is novel; consistent improvements across LLaVA-1.5, LLaVA-NeXT, and Video-LLaVA on 8+ benchmarks demonstrate broad applicability

**Weaknesses:**
- Calibration constant c=2 lacks theoretical justification and sensitivity analysis; claim about retaining original position indices assumes grid-aligned layouts that break for non-rectangular crops
- Hyperparameter gamma=0.5 selected via grid search without discussion of generalization; Appendix B shows calibration degrades performance for reindexed positions (91.0% vs 92.2% baseline); this limitation is buried rather than flagged in the main text
- Writing contains redundant phrases and awkward constructions; Table 6 lacks rigorous analysis linking median position to RoPE's rotation periodicity
- Efficiency gains somewhat overstated; O(N_vis^2) correlation computation ignored in latency terms for high-resolution images with N_vis > 1000
- Core mechanism parallels techniques in long-context LLM literature (ALiBi, xPos) without adequate situating; distinction from DivPrune's diversity criterion is subtle
- Gap between text-aware (SparseVLM 98.1%) and text-agnostic methods is small; calibration to text-aware baselines not attempted; algorithmic details are scattered across sections without consolidated pseudocode

---

> ### Author Rebuttal · Authors · 2026-03-31
>
> # Response to Reviewer BtE8
> We sincerely appreciate your thoughtful and constructive feedback. Below, we address your concerns one by one.
>
> ---
> > **Q1. Hyperparameter $c$**
> - We have set $c=2$ not only to keep the calibration term positive, but also to guarantee that the pure impact of the cluster size $s_n$ is preserved when a query attends to its own position. Specifically, at a distance of zero ($|m-n|=0$), the positional decay effect should disappear, requiring the calibration term to be $\log s_n$. Applying this to our formulation $\log s_n (c - D(|m-n|))$, since $D(0)=1$, the term simplifies to $\log s_n (c-1)$. To equate this with $\log s_n$, $c$ must be $2$. We conduct an ablation study for $c$ using LLaVA-1.5-7B with the HoloV baseline ($n_{vis}=64$). As shown in the table below, performance peaks at exactly $c=2.0$ across both the GQA and MME benchmarks, supporting our formulation.
>
> | $c$ | 1.25 | 1.50 | 1.75 | 2.00 | 2.25 | 2.50 | 2.75 | 3.00 |
> | :--- | :---: | :---: | :---: | :---: | :---: | :---: | :---: | :---: |
> | GQA | 58.0 | 58.7 | 58.7 | **59.0** | **59.0** | **59.0** | **59.0** | **59.0** |
> | MME | 1678 | 1764 | 1764 | **1806** | 1781 | 1788 | 1778 | 1775 |
> ---
> > **Q2. Comparison with the position of merged tokens**
> - We have verified this by examining the "First" anchor. If the advantage of the "Median" anchor came from a larger calibration term $(c - D(|m-n|))$, the "First" anchor would yield the highest performance due to its maximum distance offset. However, Table 6 shows that "First" achieves lower accuracy than "Median". This demonstrates that the performance gain is not a mathematical artifact of the attention calibration. Rather, it indicates that the "Median" is a more suitable proxy for the spatial location of the merged tokens.
> ---
> > **Q3. Scalability and latency overhead of anchor token selection for large $N_{vis}$**
> - We measure the wall-clock latencies, including total generation and prefill times using LLaVA-Next-7B, which processes high-resolution images with $N_{vis} > 2000$. The results are summarized in the table below. Integrating our framework into HoloV (160 tokens) incurs a negligible latency overhead while recovering substantial accuracy. Moreover, HoloV (160) + ours achieves the exact same accuracy as HoloV (320) but shows significantly lower latency (a 3.42x vs. 3.04x total speedup).
>
> | Method | Total Time | Speedup $\uparrow$ | Prefill Time | Speedup $\uparrow$ | Acc. | $\Delta$ Acc. |
> | :--- | :--- | :---: | :--- | :---: | :---: | :---: |
> | LLaVA-Next-7B | 15m 54s | 1.00x | 512.1 ms | 1.00x | 64.2 | 0.0 |
> | HoloV (320 tokens) | 5m 14s | 3.04x | 139.7 ms | 3.67x | 59.7 | -3.55% |
> | HoloV (160 tokens) | 4m 21s | 3.66x | 107.3 ms | 4.77x | 57.2 | -7.59% |
> | **+ ours (160 tokens)**| **4m 39s** | **3.42x** | **111.1 ms** | **4.61x** | **59.7** | **-3.55%** |
> ---
> > **Q4. Application to text-aware baselines**
> - Applying our framework to text-aware baselines is fully feasible. We evaluate the text-aware baselines integrated with our framework across the same benchmarks and at the same retention ratios as in Table 1.
>
> | Method | Average (%) | GQA | MMB | MME | POPE | SQA$^{IMG}$ | VQA$^{V2}$ | VQA$^{Text}$ | SEED |
> | :--- | :---: | :---: | :---: | :---: | :---: | :---: | :---: | :---: | :---: |
> | **Upper Bound, 576 Tokens (100%)** | | | | | | | | | |
> | LLaVA-1.5-7B | 100.0 | 61.9 | 64.6 | 1862 | 85.9 | 69.5 | 78.5 | 58.2 | 58.6 |
> | **Retain 192 Tokens (33.3%)** | | | | | | | | | |
> | FastV | 96.0 | 57.1 | 64.4 | 1821 | 75.8 | 68.9 | 74.7 | 57.8 | 56.3 |
> | + ours | **99.0** | 60.8 | 64.0 | 1853 | 86.0 | 68.4 | 77.4 | 57.4 | 58.1 |
> | PDrop | 96.8 | 58.0 | 62.9 | 1790 | 84.0 | 68.9 | 76.0 | 57.1 | 55.8 |
> | + ours | **99.1** | 61.0 | 64.1 | 1829 | 86.1 | 68.8 | 78.0 | 57.4 | 58.4 |
> | SparseVLM | 98.1 | 59.5 | 64.2 | 1782 | 85.4 | 68.7 | 77.0 | 57.7 | 57.3 |
> | + ours | **99.4** | 61.4 | 64.1 | 1831 | 86.5 | 69.1 | 78.1 | 57.6 | 58.5 |
> | **Retain 64 Tokens (11.1%)** | | | | | | | | | |
> | FastV | 75.2 | 46.4 | 51.4 | 1284 | 36.1 | 69.9 | 56.2 | 51.6 | 43.9 |
> | + ours | **89.9** | 54.1 | 61.2 | 1621 | 75.9 | 66.4 | 69.0 | 51.5 | 52.8 |
> | PDrop | 80.9 | 49.0 | 55.6 | 1404 | 55.5 | 69.5 | 60.1 | 50.6 | 46.1 |
> | + ours | **91.7** | 54.9 | 61.6 | 1674 | 78.2 | 67.6 | 71.1 | 52.3 | 53.5 |
> | SparseVLM | 90.6 | 53.8 | 60.2 | 1591 | 77.5 | 69.7 | 70.3 | 53.5 | 51.2 |
> | + ours | **94.5** | 57.1 | 61.9 | 1721 | 83.4 | 68.6 | 73.6 | 53.4 | 55.3 |
> ---
> > **Q5. Excessive attention to background regions and failure case**
> - We agree that the tokens in the background regions may exhibit excessive attention weights. This stems from the token size term $s_n$, which estimates the attention weights of the pre-merged tokens by multiplying the attention weight of the merged token. This estimation error is an inherent limitation of the token merging process. We believe that resolving this issue remains an important direction for future research in token merging.

---

> > ### Author Rebuttal · Reviewer_BtE8 · 2026-04-03
> >
> > Thanks for the authors detailed rebuttal, my concerns have been fully resolved.

---

> > > ### Author Response · Authors · 2026-04-03
> > >
> > > We sincerely appreciate your valuable feedback and the time you dedicated to reviewing our work.
> > > We will incorporate the results presented in our response into the revised manuscript.

---

### Official Review · Reviewer_5BCW · 2026-03-09

**Soundness:** 3
**Presentation:** 3
**Significance:** 3
**Originality:** 3
**Overall Recommendation:** 4
**Confidence:** 3

**Summary:**

This paper proposes a visual token reduction framework for MLLMs that corrects attention and position distortions introduced by pruning and merging. It calibrates attention weights using relative-distance modulation while keeping original positional indices, and improves token merging via a more representative and distinctive anchor selection, yielding consistent accuracy gains across benchmarks with minimal efficiency loss.

**Compliance With Llm Reviewing Policy:**

Affirmed.

**Final Justification:**

My concerns have been adequately addressed.

**Key Questions For Authors:**

See weakness

**Strengths And Weaknesses:**

Strength:
1. The paper raises an interesting motivation that current VTR methods overlook attentional and positional distortions when reducing visual tokens.
2. The experiments effectively validate the proposed calibration method, demonstrating consistent improvements across various VTR baselines.
3. The paper is well-written and easy to follow.

Weakness:
1. It is unclear whether the proposed calibration and anchor selection are compatible with layer-wise, adaptive pruning methods such as FlowCut [1], which estimate token importance using multiple signals across layers.
2. The paper does not evaluate the method on Qwen-VL series models. Given their popularity, it would be helpful to verify whether the proposed calibration generalizes to this model family.
3. The method requires redistributing attention weights. It is unclear whether this is compatible with FlashAttention in practice, since FlashAttention typically does not expose the full attention map.
4. The paper mainly focuses on accuracy under token reduction, but the practical speedup remains unclear. Reporting end-to-end latency including prefill time would better support the efficiency claims.

[1] FlowCut: Rethinking Redundancy via Information Flow for Efficient Vision-Language Models

---

> ### Author Rebuttal · Authors · 2026-03-31
>
> # Response to Reviewer 5BCW
> We sincerely appreciate your thoughtful and constructive feedback. Below, we address your concerns one by one.
>
> ---
> > **W1. Compatibility with layer-wise and adaptive pruning methods such as FlowCut**
> - Our proposed framework can be integrated with layer-wise, adaptive pruning methods such as FlowCut. Although FlowCut estimates token importance using multiple signals across layers, this multi-faceted criterion ultimately reduces to a single scalar importance score for each token at a given layer. Specifically, at any intermediate pruning stage $i$ designed to retain $n_{vis,i}$ tokens, we can directly apply our hybrid reduction strategy: retaining the top $\gamma \cdot n_{vis,i}$ tokens via pruning based on the FlowCut scores, and constructing the remaining $(1-\gamma) \cdot n_{vis,i}$ tokens via merging using our distinctive anchor selection.
>
> - We show in the table below the results of FlowCut and its integration with our framework on eight benchmarks at two retention ratios ($n_{vis}=64$ and $128$). Considering that FlowCut involves multiple pruning stages, we additionally experiment with higher pruning ratios ($\gamma=0.75$ and $0.9$) to mitigate potential feature distortion during iterative token merging. Our framework consistently improves the performance of FlowCut across all settings. It is worth noting that $\gamma=0.75$ yields the best overall performance, demonstrating that a balanced ratio is ideal for multi-stage adaptive methods. We will include these results and the integration methodology for layer-wise and adaptive pruning methods such as FlowCut in the revised version.
>
> | Method | Average (%) | GQA | MMB | MME | POPE | SQA$^{IMG}$ | VQA$^{V2}$ | VQA$^{Text}$ | SEED |
> | :--- | :---: | :---: | :---: | :---: | :---: | :---: | :---: | :---: | :---: |
> | LLaVA-1.5-7B (576 tokens) | 100.0 | 61.9 | 64.6 | 1862 | 85.9 | 69.5 | 78.5 | 58.2 | 58.6 |
> | FlowCut (128 tokens) | 97.0 | 58.5 | 62.1 | 1792 | 85.2 | 68.6 | 76.0 | 57.3 | 56.2 |
> | + ours ($\gamma=0.5$) | 97.2 | 60.6 | 63.3 | 1736 | 86.3 | 66.9 | 77.3 | 55.7 | 57.3 |
> | + ours ($\gamma=0.75$) | **97.7** | 60.4 | 63.2 | 1764 | 86.3 | 68.3 | 77.4 | 55.8 | 57.5 |
> | + ours ($\gamma=0.9$) | 97.5 | 59.8 | 62.5 | 1819 | 85.3 | 68.3 | 77.2 | 55.6 | 57.3 |
> | FlowCut (64 tokens) | 93.7 | 55.6 | 60.8 | 1744 | 80.2 | 69.1 | 72.8 | 55.6 | 53.5 |
> | + ours ($\gamma=0.5$) | 95.3 | 58.9 | 62.2 | 1690 | 84.8 | 67.9 | 75.6 | 53.3 | 56.2 |
> | + ours ($\gamma=0.75$) | **95.7** | 58.7 | 63.1 | 1749 | 83.8 | 68.0 | 75.5 | 53.9 | 55.5 |
> | + ours ($\gamma=0.9$) | 94.2 | 56.9 | 61.4 | 1746 | 81.9 | 67.8 | 74.3 | 54.0 | 54.4 |
>
> ---
> > **W2. Attention calibration for Qwen2.5-VL with M-RoPE**
> - We have extended the attention calibration to M-RoPE with an experiment on Qwen2.5-VL. Please refer to our response to W1 from Reviewer 6H88.
>
> ---
> > **W3. Compatibility with FlashAttention**
> - The concern stems from the assumption that our method requires a post-softmax redistribution of the full attention map. However, our attention calibration operates as a pre-softmax logit bias. Specifically, our calibrated attention simplifies to $\hat{A}\_{m,n} = \text{softmax}(z\_{m,n}+\log s\_n (c - D(|m-n|)))$, where $z\_{m,n}$ is the standard attention logit for query position $m$ and key position $n$, and the second term is a deterministic calibration term depending on $m$ and $n$. Since it is applied prior to the softmax operation, this calibration is compatible with FlashAttention's block-wise tiling mechanism.
>
> ---
> > **W4. Practical speedup results including end-to-end latency**
> - We measure the end-to-end latency including total inference time and prefill time on the GQA dataset using an 8× A5000 GPU setup. As shown in the table below, integrating our framework introduces only a marginal latency overhead compared to the baseline, while yielding substantial accuracy improvements.
>
> | Method | Total Time | $\Delta \uparrow$ (Speedup) | Prefill Time | $\Delta \uparrow$ (Speedup) | Acc. | $\Delta \uparrow$ (Acc.) |
> | :--- | :---: | :---: | :---: | :---: | :---: | :---: |
> | LLaVA-1.5-7B | 6m05s | 1.00x | 227.8ms | 1.00x | 61.9 | 0% |
> | HoloV (192 tokens) | 4m23s | 1.39x | 178.2ms | 1.28x | 58.6 | -5.33% |
> | + ours | 4m28s | 1.36x | 181.4ms | 1.26x | 61.0 | -1.45% |
> | HoloV (128 tokens) | 3m58s | 1.53x | 158.2ms | 1.44x | 57.5 | -7.11% |
> | + ours | 4m02s | 1.51x | 164.0ms | 1.39x | 60.8 | -1.78% |
> | HoloV (64 tokens) | 3m39s | 1.67x | 145.6ms | 1.56x | 55.1 | -10.99% |
> | + ours | 3m43s | 1.64x | 151.1ms | 1.51x | 59.0 | -4.68% |

---

> > ### Author Rebuttal · Reviewer_5BCW · 2026-04-02
> >
> > I appreciate the authors' detailed response to my concerns. The issues raised in W1–W4 have been satisfactorily addressed. I recommend that the authors incorporate the relevant experiments and explanations into the revised manuscript to strengthen the presentation. Accordingly, I have raised my score from 3 to 4.

---

> > > ### Author Response · Authors · 2026-04-02
> > >
> > > We sincerely appreciate your time during the review process and the raised evaluation. We agree with your recommendation to strengthen the manuscript. We will incorporate the additional results and clarifications from our response to W1-W4 into the revised version.

---

### Official Review · Reviewer_3faY · 2026-03-12

**Soundness:** 2
**Presentation:** 2
**Significance:** 3
**Originality:** 3
**Overall Recommendation:** 4
**Confidence:** 4

**Summary:**

This paper focuses on improving visual token reduction and addresses the positional and attentional distortion problem brought by RoPE’s distance bias and softmax normalization. It proposes a training-free attention calibration to rectify the distortions and introduces a distinctive anchor token selection strategy to mitigate information loss. Extensive experiments across different benchmarks demonstrate consistent improvements.

**Compliance With Llm Reviewing Policy:**

Affirmed.

**Final Justification:**

The supplemented results and analysis have addressed my concerns, and I have raised my score accordingly.

**Key Questions For Authors:**

1. In line 098, “we propose a VTR framework that integrates pruning and merging to rectify distortions in attentional and positional information.” This sentence claims that pruning and merging are designed to rectify distortions. However, the merging side improvement, distinctive anchor selection, is designed for mitigating information loss. Is there any logical error in this sentence?
2. How was the selection made for how many tokens to retain in the experiment? Why did the authors choose these configurations? Why were more tokens, such as over 50%, not considered?
3. Table 5-8 lacks necessary explanations. How were these numbers obtained, and on what data? For Table 4, why do the authors choose HoloV with $n_{vis} = 64$ for the ablation study?
4. Whether the proposed method will work on tasks requiring fine-grained visual perception, like optical character recognition (OCR)?

**Limitations:**

This paper lacks a limitation discussion. To improve, it should discuss in which circumstance will the proposed method falls short. For example, the experiments are not conducted on newer backbones like Qwen2.5-VL, and attention calibration relies on RoPE.

**Strengths And Weaknesses:**

### Strengths

1. The attention calibration is well-motivated. It starts with an analysis regarding attentional and positional distortions and verifies their widespread existence, followed by an attention calibration to mitigate them.
2. This paper validates on 4 different baselines across 8 benchmarks, and achieves consistent improvements, demonstrating the effectiveness of the proposed framework.

### Weaknesses

1. (Main issue) The paper lacks an analysis of information loss in the token merging process.
    1. Unlike the well-motivated attention calibration, the design of the anchor token selection is confusing. To improve, the paper should include a proof of information loss, a method for measuring it, and an analysis of how the proposed anchor selection reduces information loss (not just performance gains).
    2. In the introduction part, the main motivation part, which is analyzed with an illustration in Figure 1, does not include any reason for the introduction of the anchor selection strategy, which makes it hard to understand.
2. Some claims lack verification. For example, in line 047, “Due to the exponential nature of softmax, the redistribution amplifies the remaining tokens that possessed large attention weights.” This claim, though explained visually in illustration (Figure 1), remains intuitive and is not proved theoretically or experimentally.
3. The attention calibration is designed for RoPE, which will limit the application scope of the proposed method and should at least be pointed out in the Limitations section.

---

> ### Author Rebuttal · Authors · 2026-03-31
>
> # Response to Reviewer 3faY
> We sincerely appreciate your thoughtful and constructive feedback. Below, we address your concerns one by one.
>
> > **W1. Lack of analysis in the token merging**
> - We adopt the Hausdorff distance [1, 2] to quantify information loss. The standard Hausdorff distance minimizes distance to the nearest anchor but treats all anchors equally, ignoring the broader coverage of merged tokens. Given the set of original visual tokens $X$ and the set of merged anchor tokens $Y$, a merged anchor $y_j \in Y$ is not just a single point, but a proxy representing a cluster of $s_j$ tokens. Since a larger cluster inherently covers a wider semantic region, the distance penalty to $y_j$ should be discounted proportionally to its cluster size. To formulate this, we define a weighted Hausdorff distance that divides the cosine distance by $s_j$:
>
> $$h_w(X, Y) = \max_{x \in X} \min_{y_j \in Y} \frac{1 - \cos(x, y_j)}{s_j}$$
>
> - We measure $h_w$ on four baselines under three configurations: Pruning only, Pruning + Merging with uniform anchors, and our distinctive anchors. These values are averaged over 100 non-overlapping images from the GQA dataset using LLaVA-1.5-7B. The results show that pruning alone yields the highest information loss. While uniform merging reduces this loss, our distinctive anchors achieve the lowest $h_w$ across all baselines.
>
> | Baseline | Pruning Only | P + M (Uniform) | **P + M (Distinctive)** |
> | :--- | :---: | :---: | :---: |
> | VisionZip | 0.7168 | 0.0169 | **0.0077** |
> | DivPrune | 0.2889 | 0.0177 | **0.0078** |
> | VisPruner | 0.4590 | 0.0169 | **0.0078** |
> | HoloV | 0.6701 | 0.0167 | **0.0076** |
>
> [1] Comparing images using the Hausdorff distance, TPAMI 2002.
> [2] DivPrune: Diversity-based Visual Token Pruning for Large Multimodal Models, CVPR 2025
>
> ---
> > **W2. Lack of verification for the exponential nature of softmax**
>
> Let $S$ be the original set of all tokens ($|S| = N_{sys} + N_{vis} + N_{txt}$) and $P$ be the set of pruned visual tokens. For any token $i$, let $z_i$ be its attention logit. The original attention weight is:
> $$a_i = \frac{\exp(z_i)}{Z}, \quad \text{where} \quad Z = \sum_{j \in S} \exp(z_j)$$
> After pruning the visual tokens from $N_{vis}$ to $n_{vis}$, the partition function reduces to $Z' = Z - \sum_{k \in P} \exp(z_k)$. The new attention weight for a remaining token $i \in S \setminus P$ is as follows:
> $$a'_i = \frac{\exp(z_i)}{Z'}$$
> The amplification in the attention weight for token $i$ is:
> $$\Delta a_i = a'_i - a_i = \exp(z_i) \left( \frac{1}{Z'} - \frac{1}{Z} \right) = a_i \left( \frac{Z - Z'}{Z'} \right)$$
> Since $Z > Z'$, the term $\frac{Z - Z'}{Z'}$ is a positive constant across all remaining tokens. This derivation demonstrates that the redistributed probability mass from the pruned tokens is not shared uniformly. Instead, the absolute increase $\Delta a_i$ is proportional to its original weight $a_i$. That is, tokens that originally possessed larger attention weights absorb a larger portion of the redistributed weights.
>
> ---
> > **W3. Attention calibration for Qwen2.5-VL with M-RoPE**
> - We have extended the attention calibration to M-RoPE with an experiment on Qwen2.5-VL. Please refer to our response to W1 from Reviewer 6H88.
> ---
> > **Q1. Claim in Line 098**
> - We will correct the imprecision in Line 098 in the revised version to reflect each component's distinct purpose.
> ---
> > **Q2. Justification for the chosen token retention ratios**
> - We have selected retention ratios to align with baseline methods for fair comparisons. We have excluded higher retention rates since the baselines already achieve nearly lossless performance at 33.3% (e.g., >95%).
> ---
> > **Q3. Experimental details for Tables 5-8 and rationale for Table 4**
> - For Tables 5, 6, and 8, the results have measured by isolating the impact of each component by fixing the others in Table 1 with $n_{vis}=64$.
> - We have selected $n_{vis}=64$ to observe performance variations under a challenging retention ratio and HoloV as a baseline since it represents the most recent SOTA method.
> ---
> > **Q4. Experiment on fine-grained visual perception task**
> - To verify our method's effectiveness on tasks requiring fine-grained visual perception, we conduct experiments on the OCRBench dataset. As shown in the table below, integrating our framework with 4 VTR baselines yields performance improvements across different retention ratios ($n_{vis}=192, 128,$ and $64$), with a minor exception under the extreme compression setting (VisPruner at $n_{vis}=64$).
>
> | Retained Tokens | Method | VisionZip | DivPrune | VisPruner | HoloV |
> | :--- | :--- | :---: | :---: | :---: | :---: |
> | **$n_{vis}=192$** | Vanilla | 286 | 281 | 295 | **295** |
> | | + ours | **290** | **293** | **298** | **295** |
> | **$n_{vis}=128$** | Vanilla | 285 | 264 | 290 | 288 |
> | | + ours | **287** | **288** | **294** | **301** |
> | **$n_{vis}=64$** | Vanilla | 258 | 257 | **287** | 279 |
> | | + ours | **269** | **267** | 273 | **283** |

---

> > ### Author Rebuttal · Reviewer_3faY · 2026-04-02
> >
> > I thank the authors for their detailed response.
> >
> > Regarding W1, I suggest **rewriting the introduction** to prevent token merging from appearing directly and causing confusion for readers. For W2, the clarifications should be integrated into the paper. Similarly, for Q3, more detailed experimental settings are necessary in the manuscript to ensure **reproducibility**. The experiments in Q4 should also be included in the main results (fine-grained visual perception is not a negligible task but a **major application direction**).
> >
> > I have raised my score accordingly.

---

> > > ### Author Response · Authors · 2026-04-02
> > >
> > > We sincerely appreciate your time during the review process and for raising your score. We agree with your final suggestions for improving the manuscript. In the revised version, we will ensure the following:
> > >
> > > - [W1] We will revise the introduction to prevent confusion regarding token merging.
> > > - [W2] We will include the verification of “the exponential nature of softmax”.
> > > - [Q3] We will elaborate on the experimental details and release our code to ensure reproducibility.
> > > - [Q4] We will include the OCRBench experiments in the main text.

---

### Official Review · Reviewer_6H88 · 2026-03-12

**Soundness:** 4
**Presentation:** 3
**Significance:** 3
**Originality:** 4
**Overall Recommendation:** 5
**Confidence:** 4

**Summary:**

The manuscript addresses a reseach gap in existing visual token reduction (VTR) methods, which is attention distortion arising from (1) re-indexing that disrupts spatial relationships between tokens and (2) rigid anchor token selection that overlooks information loss. To address these, the paper proposes (1) a relative distance-aware attention calibration mechanism that counteracts RoPE's long-term decay term and (2) distinctive anchor token selection strategy that jointly maximizes representativeness and discriminativeness during merging based on inter-token correlation significancy and redundancy, respectively. The proposed framework is evaluated using diverse MLLM benchmarks and foundational models, and compared with diverse SOTA VTR methods, showing consistent improvements across various settings.

**Compliance With Llm Reviewing Policy:**

Affirmed.

**Final Justification:**

The rebuttal has satisfactorily addressed my concerns.

**Key Questions For Authors:**

NA

**Limitations:**

Yes

**Strengths And Weaknesses:**

### Strengths
- [s1]: Research gap is well-motivated and analysis is formally grounded for RoPE's attentional distortion under VTR. This analysis constitutes a novel contribution to the understanding of VTR-PE interactions.
- [s2]: Proposed solutions are compelling and show consistent and broad empirical improvements; they consistently improves all baselines across all MLLM benchmark and architectures. This breadth of improvement strengthens the claim of generalizability.
- [s3]: Moreover, their solution does not come at cost of computation. There is only 0.074-0.136% of total inference FLOPs overhead. Figure 4 nicely show accuracy-latency trade-off curve confirming that the latency increase is marginal relative to the accuracy gain.
- [s4]: Authors conducted comprehensive ablation study showing contribution of each component in a well-structured way. Notably, it reveals that applying merging with position retaining but without attention calibration is worse than the pure-pruning baseline, which exposes a failure mode and justifies the joint design.

### Weaknesses
- [w1]: The proposed calibration is motivated by and derived for standard RoPE. However, M-RoPE, which is used in Qwen2.5-VL, extends RoPE to handle 2D spatial positions differently. It remains unclear whether the long-term decay analysis in Section 3 carries over to M-RoPE without modification, and a brief discussion or experiment on this would strengthen the paper's positioning.
- [w2]: The framework is never applied to text-aware baselines. The paper mentions that text-agnostic methods are preferred for efficiency, though this does not really clarify whether applying the calibration and anchor selection to text-aware methods is technically infeasible or simply untried.
- [w3]: Two performance regressions on VQAText is not acknowledged in the paper. In Table 1, at 192-token retention, VisionZip + ours scores 54.9 on VQAText, below the VisionZip baseline of 55.8 (−0.9). Similarly in Table 2 at 160 tokens for LLaVA-Next-7B, the score drops from 54.5 to 51.0. These are the only cases in the main results where the proposed method underperforms a baseline, and they are never discussed. This pattern may suggests the attention calibration may occasionally over-suppress text-relevant tokens or under-attend short-range visual features, which seems worth investigating.
- [w4]: In Table 4, Row 6 at 92.9% is lower than Row 5 at 93.5%, which is not acknowledged in the paper.

---

> ### Author Rebuttal · Authors · 2026-03-31
>
> # Response to Reviewer 6H88
> We sincerely appreciate your thoughtful and constructive feedback. Below, we address your concerns one by one.
>
> ---
> > **W1. Attention calibration for Qwen2.5-VL with M-RoPE**
>
> In standard 1D RoPE, our decay function ${D}$ (Eq. 3) averages cosine terms across all frequency dimensions. In M-RoPE, the half-head dimension $d_h/2$ is partitioned into $G$ independent axes (e.g., temporal, height, width). Each axis $g \in \{1, \dots, G\}$ is allocated $d_g$ dimensions and has its own positional distance $\Delta_g$. To extend our attention calibration to M-RoPE, we generalize the standard 1D RoPE formulation into a weighted sum of per-axis decay functions:
>
> $D_{M\text{-}RoPE}=\sum_{g=1}^{G} w_g D_g(\Delta_g)$
>
> where $D_g(\Delta_g) = \frac{1}{|\Omega_g|} \sum_{j \in \Omega_g} \cos(\Delta_g \cdot \theta_j)$ computes the specific decay for axis $g$. $\Omega_g$ represents the set of frequency indices assigned to axis $g$ ($|\Omega_g| = d_g$), and $\theta_j$ is the rotation frequency for index $j$. The weight $w_g = 2d_g / d_h$ denotes the proportion of dimensions allocated to axis $g$. The weight $w_g = 2d_g / d_h$ denotes the proportion of dimensions allocated to axis $g$, which guarantees $\sum w_g = 1$ and preserves the scale of the standard 1D RoPE. This formulation preserves our fundamental principle: attention inherently decays as positional distance increases, requiring a proportional calibration bias in the VTR scenario. To validate this, we evaluate Qwen2.5-VL-7B-Instruct across five benchmarks, confirming that our method consistently improves performance over the VTR baselines.
>
> | Method | Average (%) | MMB | MME | POPE | SQA | TextVQA |
> | :--- | :---: | :---: | :---: | :---: | :---: | :---: |
> | **Using all visual tokens** | | | | | | |
> | Qwen2.5-VL-7B-Instruct | 100.0 | 84.4 | 2323 | 86.7 | 77.8 | 77.7 |
> | **Retain 33.3% visual tokens** | | | | | | |
> | DivPrune (CVPR 25) | 95.9 | 79.6 | 2187 | 84.9 | 76.2 | 73.8 |
> | + ours | **97.2** | 81.1 | 2205 | 86.2 | 78.2 | 73.8 |
> | VisPruner (ICCV 25) | 95.8 | 80.2 | 2175 | 84.6 | 76.7 | 73.2 |
> | + ours | **97.4** | 82.0 | 2211 | 86.0 | 78.6 | 73.3 |
> | HoloV (NeurIPS 25) | 93.6 | 78.2 | 2121 | 83.6 | 75.9 | 70.0 |
> | + ours | **96.4** | 81.0 | 2193 | 85.7 | 78.1 | 71.9 |
> | **Retain 11.1% visual tokens** | | | | | | |
> | DivPrune (CVPR 25) | 87.5 | 71.1 | 1961 | 79.7 | 72.7 | 64.9 |
> | + ours | **89.8** | 74.5 | 1998 | 82.3 | 76.3 | 63.6 |
> | VisPruner (ICCV 25) | 86.9 | 69.8 | 2137 | 77.0 | 73.1 | 59.8 |
> | + ours | **88.5** | 74.4 | 1980 | 80.8 | 77.1 | 59.6 |
> | HoloV (NeurIPS 25) | 83.5 | 72.3 | 1834 | 77.6 | 70.8 | 56.4 |
> | + ours | **88.4** | 76.3 | 1992 | 81.3 | 75.4 | 58.3 |
>
> ---
> > **W2. Application to text-aware baselines**
> - We have validated our framework on text-aware baselines and confirmed performance improvement, but we are unable to report the full results here due to character limits. Please refer to our response to Q4 from Reviewer BtE8.
>
> ---
> > **W3. Performance degradation in TextVQA dataset**
> - We investigate the performance drops on TextVQA, and our analysis reveals that the regression is not caused by the attention calibration over-suppressing tokens, but rather by the feature averaging during the token merging phase. Tasks like TextVQA require reading fine-grained, highly localized text from images. This critical information is often concentrated in a very small number of visual tokens. When we apply token merging, these crucial tokens are averaged with other tokens, damaging the localized high-frequency details.
>
> - To validate this, we conduct an ablation study on the pruning ratio $\gamma$ combined with our attention calibration on TextVQA with $n_{vis}=192$. As shown in the table below, when we completely bypass merging ($\gamma=1$), the performance recovers to or even exceeds the baseline. This confirms that for tasks requiring fine-grained visual perception, the hybrid strategy should adaptively avoid merging to prevent critical information loss.
>
> | Method | Baseline | $\gamma=0$ (Only Merging) | $\gamma=0.25$ | $\gamma=0.5$ | $\gamma=0.75$ | $\gamma=1$ (Only Pruning) |
> | :--- | :---: | :---: | :---: | :---: | :---: | :---: |
> | VisionZip | 55.8 | 53.6 | 54.6 | 54.9 | 54.7 | 55.8 |
> | DivPrune | 55.7 | 56.8 | 56.5 | 56.5 | 56.5 | 56.8 |
> | VisPruner | 57.7 | 56.8 | 56.9 | 57.0 | 57.0 | 57.7 |
> | HoloV | 57.2 | 56.8 | 56.9 | 57.2 | 57.3 | 57.2 |
>
> ---
> > **W4. No explanation of Row 6 in Table 4**
> - The performance drop from Row 5 to Row 6 shares the exact same mechanism as the drop from Row 1 to Row 2. As explained in Appendix B (Figure 5), applying attention calibration to reindexed sequences causes an attention imbalance by allocating excessive attention to visual tokens and ignoring textual information. We will explicitly include this discussion in the main text of the revised version.

---

> > ### Author Rebuttal · Reviewer_6H88 · 2026-04-02
> >
> > I appreciate the authors’ additional explanations and clarifications. My concerns have been fully resolved. I have raised my score accordingly.

---

> > > ### Author Response · Authors · 2026-04-03
> > >
> > > We sincerely appreciate your time and constructive feedback during the review process.
> > > We will incorporate the results presented in our response to W1-4 into the revised manuscript.

---

### Decision · Program_Chairs · 2026-04-30

**Decision:**

Accept (regular)

**Comment:**

This paper studies visual token reduction for MLLMs and identifies an important limitation of existing methods: token reduction can introduce positional and attentional distortions, which in turn harm representation quality. To address this, the paper proposes a training-free framework with two main components: a distance-aware attention calibration method to compensate for attention distortion, and a distinctive anchor selection strategy for token merging to reduce information loss. The paper tackles a timely and practically relevant problem, and the proposed method is shown to improve a range of existing reduction methods while maintaining efficiency.

Reviewers were broadly positive about this paper. The main strengths highlighted were the clear motivation, the principled analysis of distortion under visual token reduction, the simplicity and practicality of the proposed calibration framework, and the strong empirical results across multiple benchmarks, backbones, and reduction baselines. Reviewers also noted that the added computational overhead is very small relative to the accuracy gains, which strengthens the practical value of the work.

The main concerns raised in the initial reviews included the scope of the RoPE-based analysis, the treatment of token merging and information loss, generalization to newer architectures such as Qwen2.5-VL, applicability to text-aware or adaptive pruning baselines, fine-grained perception settings such as OCR/TextVQA, and clarification of efficiency and latency overhead. In my assessment, the rebuttal addressed these issues well. The authors provided additional theoretical clarification, new experiments on Qwen2.5-VL, OCRBench, FlowCut, and text-aware baselines, as well as further latency analysis and discussion of the method’s limitations. Importantly, all reviewers indicated that their concerns were fully resolved after rebuttal.

Overall, I find this to be a technically solid and practically meaningful contribution. The paper presents a clear diagnosis of an underexplored problem in visual token reduction, proposes simple but effective remedies, and supports its claims with broad empirical evidence. Given the consistently positive reviews, the strengthened rebuttal, and the fact that all the raised concerns have been fully resolved, I recommend acceptance.